# The RNA-binding profile of the splicing factor SRSF6 in immortalized human pancreatic β-cells

Maria Inês Alvelos[1], Mirko Brüggemann[2,3], FX Reymond Sutandy[4], Jonàs Juan-Mateu[1,5], Maikel Luis Colli[1], Anke Busch[4], Miguel Lopes[1], Ângela Castela[1], Annemieke Aartsma-Rus[6], Julian König[4], Kathi Zarnack[2,3], Décio L Eizirik[1,7,8]

In pancreatic β-cells, the expression of the splicing factor SRSF6 is regulated by GLIS3, a transcription factor encoded by a diabetes susceptibility gene. *SRSF6* down-regulation promotes β-cell demise through splicing dysregulation of central genes for β-cells function and survival, but how RNAs are targeted by SRSF6 remains poorly understood. Here, we define the SRSF6 binding landscape in the human pancreatic β-cell line EndoC-βH1 by integrating individual-nucleotide resolution UV cross-linking and immunoprecipitation (iCLIP) under basal conditions with RNA sequencing after *SRSF6* knockdown. We detect thousands of SRSF6 bindings sites in coding sequences. Motif analyses suggest that SRSF6 specifically recognizes a purine-rich consensus motif consisting of GAA triplets and that the number of contiguous GAA triplets correlates with increasing binding site strength. The SRSF6 positioning determines the splicing fate. In line with its role in β-cell function, we identify SRSF6 binding sites on regulated exons in several diabetes susceptibility genes. In a proof-of-principle, the splicing of the susceptibility gene *LMO7* is modulated by antisense oligonucleotides. Our present study unveils the splicing regulatory landscape of SRSF6 in immortalized human pancreatic β-cells.

## Introduction

Alternative splicing (AS) is a key co- and post-transcriptional mechanism regulating eukaryotic gene expression that determines which transcript and protein isoforms are formed under specific physiological and cellular contexts (Nilsen & Graveley, 2010; Liu et al, 2017). More than 95% of human multi-exon genes experience AS, generating two or more different isoforms per gene (Nilsen & Graveley, 2010; Barbosa-Morais et al, 2012; Merkin et al, 2012). AS is regulated by RNA-binding proteins (RBPs) and their interactions with core spliceosomal components (Sanford et al, 2008; Wahl et al, 2009). RBPs bind to pre-mRNA binding sites that act as splicing-regulatory elements either enhancing or repressing the recognition of consensus splicing sequences and spliceosome assembly, thereby determining the final splicing outcome. The function of RBPs as enhancers or repressors of splicing often depends on the positioning of their binding sites relative to the regulated splice sites (Witten & Ule, 2011). Moreover, different RBPs may work in cooperation or antagonistically to regulate overlapping sets of AS events (Matlin et al, 2005; Sanford et al, 2008; Barash et al, 2010). Mutations in regulatory factors, both spliceosomal components and RBPs, or in pre-mRNA sequences can lead to splicing dysregulation. Splicing alterations have been extensively studied in multiple tissues and splicing dysregulation has been increasingly recognized as a molecular mechanism associated with multiple human diseases (Braunschweig et al, 2013; Kelemen et al, 2013; Montes et al, 2019).

Diabetes is a chronic metabolic dysfunction characterized by deterioration and loss of the insulin-producing pancreatic β-cells, resulting in hyperglycemia and long-term complications (Weir & Bonner-Weir, 2013; Eizirik et al, 2020). The two main forms of diabetes are type 1 (T1D) and type 2 (T2D) diabetes; they are triggered by different mechanisms, that is, autoimmunity in T1D and metabolic stressors in T2D, but in both cases genetic and environmental interactions prompt the failure and/or loss of the insulin-producing pancreatic β-cells leading to chronic hyperglycemia (Dooley et al, 2016; Ramos-Rodríguez et al, 2019; Eizirik et al, 2020). Of note, many of the susceptibility genes for diabetes act at the β-cell level, and risk single nucleotide polymorphisms (SNPs) affect both coding and noncoding regions (Ramos-Rodríguez et al, 2019; Colli et al, 2020; Gonzalez-Moro et al, 2020). Transcriptome analysis indicates that

---

[1]ULB Center for Diabetes Research, Medical Faculty, Université Libre de Bruxelles (ULB), Brussels, Belgium  [2]Buchman Institute for Molecular Life Sciences (BMLS), Goethe University Frankfurt, Frankfurt am Main, Germany  [3]Faculty of Biological Sciences, Goethe University Frankfurt, Frankfurt am Main, Germany  [4]Institute of Molecular Biology gGmbH, Mainz, Germany  [5]Centre for Genomic Regulation, The Barcelona Institute of Science and Technology, Barcelona, Spain  [6]Leiden University Medical Center, Leiden, The Netherlands  [7]Welbio, Medical Faculty, Université Libre de Bruxelles (ULB), Brussels, Belgium  [8]Indiana Biosciences Research Institute, Indianapolis, IN, USA

Correspondence: mardeoli@ulb.ac.be; kathi.zarnack@bmls.de; deizirik@ulb.ac.be
FX Reymond Sutandy's present address is Institute of Biochemistry II, Goethe University Frankfurt, Frankfurt am Main, Germany.
Maria Inês Alvelos and Mirko Brüggemann are shared first authors

---

dysregulated splicing is one of the mechanisms that contribute to pancreatic β-cell dysfunction and death in diabetes (Eizirik et al, 2012). Indeed, exposure of human β-cells to pro-inflammatory cytokines or the metabolic stressor palmitate, in vitro models of β-cell stress in T1D or T2D, respectively, induce major changes in the AS repertoire, affecting the splicing of key genes for β-cell function and survival (Eizirik et al, 2012; Cnop et al, 2014; Colli et al, 2020). Nevertheless, knowledge on the role of AS in pancreatic β-cell dysfunction and death in diabetes is still in its infancy (reviewed in Juan-Mateu et al [2016] and Alvelos et al [2018]).

We have previously shown that down-regulation of *GLIS3*, a susceptibility gene for T1D and T2D development that is also causal for monogenic forms of the disease (Taha et al, 2003; Senée et al, 2006; Dimitri et al, 2011), decreases the expression of the splicing factor serine and arginine rich 6 splicing factor (SRSF6, also known as SRp55), a serine/arginine (SR) protein family member (Nogueira et al, 2013; Juan-Mateu et al, 2018). SR proteins are a highly conserved family of splicing regulators, with a central role in both constitutive and AS (Graveley, 2000; Long & Cáceres, 2009). They are ubiquitously expressed but regulate splicing in a cell type- and concentration-dependent manner, by contributing to splice site selection (Hanamura et al, 1998; Bourgeois et al, 2004; Chen & Manley, 2009). *SRSF6* has been described as a proto-oncogene and its aberrant expression contributes to skin, lung, breast, and colorectal cancer (Cohen-Eliav et al, 2013; Jensen et al, 2014; Silipo et al, 2015; Dvinge et al, 2016; Park et al, 2019). *SRSF6* is highly expressed in both the pancreatic β-cell line EndoC-βH1 and in human islets of Langerhans, and its down-regulation has a major impact on β-cell function and survival (Juan-Mateu et al, 2018). By performing RNA sequencing (RNA-seq) following *SRSF6* knockdown (KD) in EndoC-βH1 cells, we have previously shown that SRSF6 modulates the splicing of genes involved in different biological processes, such as β-cell survival, insulin secretion, and c-Jun N-terminal kinase (JNK) signaling (Juan-Mateu et al, 2018). Thus, SRSF6 down-regulation leads to the generation of alternatively spliced isoforms that promote EndoC-βH1 cell and primary human islet cell dysfunction and death (Juan-Mateu et al, 2018). These experiments, however, did not allow us to discriminate between direct and indirect effects of SRSF6 on the target transcripts.

In the present study, we integrated transcriptome-wide SRSF6 binding maps from individual-nucleotide resolution UV cross-linking and immunoprecipitation (iCLIP) with SRSF6-associated AS profiles to identify the mechanisms of SRSF6-mediated splicing regulation in the human pancreatic β-cell line EndoC-βH1. This combined approach unveiled how SRSF6 specifically recognizes its binding sites and promotes exon inclusion, thereby regulating the splicing of diabetes susceptibility genes. These findings disclose a novel layer of regulation in the genetic predisposition to diabetes.

# Results

### Optimized conditions for SRSF6 iCLIP in the human β-cell line EndoC-βH1

To presently apply individual-nucleotide UV cross-linking and immunoprecipitation (iCLIP), we first adapted the conditions for

EndoC-βH1 cells. The EndoC-βH1 cell line is an established in vitro model that resembles human islet β-cells in terms of gene expression, functionality and metabolic properties (Scharfmann et al, 2016; Tsonkova et al, 2018). *SRSF6* gene expression in EndoC-βH1 cells was comparable with HeLa and HEK293 cells (Fig S1A). We confirmed the specificity of the primary antibody in EndoC-βH1 cells by comparing it to a *SRSF6* knockdown (KD) using a previously validated siRNA which leads to ≥50% SRSF6 depletion at mRNA and protein level (Juan-Mateu et al, 2018). Western blot analysis confirmed the detection of only one band in wild-type cells which was diminished by *SRSF6* KD (Fig S1B). Next, we optimized the UV cross-linking conditions. HeLa cells were used as a positive control. In both cell lines, the exposure to UV light (254 nm, 150 mJ/cm$^2$) induced the appearance of cross-linked SRSF6–RNA complexes (Fig 1A, lane 1 and 3). A control treatment with high RNase concentration confirmed the presence of a single band, underlining the specificity of the immunoprecipitation (Fig 1A, lane 2, Fig S1C) However, we retrieved considerably less cross-linked material from EndoC-βH1 cells compared with the same amount of HeLa cells. Doubling the UV energy improved the yield from EndoC-βH1 cells (Fig 1A, lane 5), this approach being more effective than increasing the number of irradiated cells (Fig 1A, lane 4) or a combination of both (Fig 1A, lane 6). Based on these results, we performed the subsequent iCLIP experiments using $8 \times 10^6$ cells and a UV irradiation energy of 300 mJ/cm$^2$ (Fig S1D).

### SRSF6 shows sequence-specific binding on thousands of protein-coding genes

To map the binding of SRSF6 in human pancreatic β-cells, we performed iCLIP for SRSF6 from EndoC-βH1 cells in four independent replicates (Fig S1D). We obtained a total of 68,449,054 cross-link events for SRSF6. These cumulated into 185,266 reproducible binding sites that were assigned to 8,533 genes. 93% of the binding sites occurred in protein-coding genes (Fig S1E). The bound genes were associated with a broad range of different functionalities, including DNA repair, RNA splicing and cell cycle progression (Fig 1B). Within the transcript, we observed a similar fraction of SRSF6 binding sites in introns (48%) and coding sequences (CDS, 40%; Fig 1C and D). In relatin to the relative size of these regions, this documented a strong preference for SRSF6 binding in exons (Fig 1C, bottom), as exemplified in the *CCDC50* gene (Fig 1D).

To investigate the RNA binding preferences of SRSF6, we analyzed the sequence composition in a 49-nt window around SRSF6 binding sites. We found that binding sites in CDS frequently displayed AG-rich pentamers, most prominently GAAGA, AGAAG, and AAGAA (Fig 2A). In contrast, SRSF6 binding sites in introns were dominated by uridine-rich pentamers, most predominantly UUUUU, possibly reflecting the sequence composition in introns and the UV cross-linking bias inherent for iCLIP experiments (characterized in detail in Sugimoto et al [2012] and Chakrabarti et al [2018]). Consistently, UUUUU was most enriched at binding site centers, whereas the AG-rich pentamers accumulated up- and downstream (Figs 1D, 2A–C, and S2A and B). Metaprofiles revealed a rising frequency of AAGAA and GAAGA from up to 100-nt upstream, which sharply dropped ~25-nt downstream of the SRSF6 binding sites,

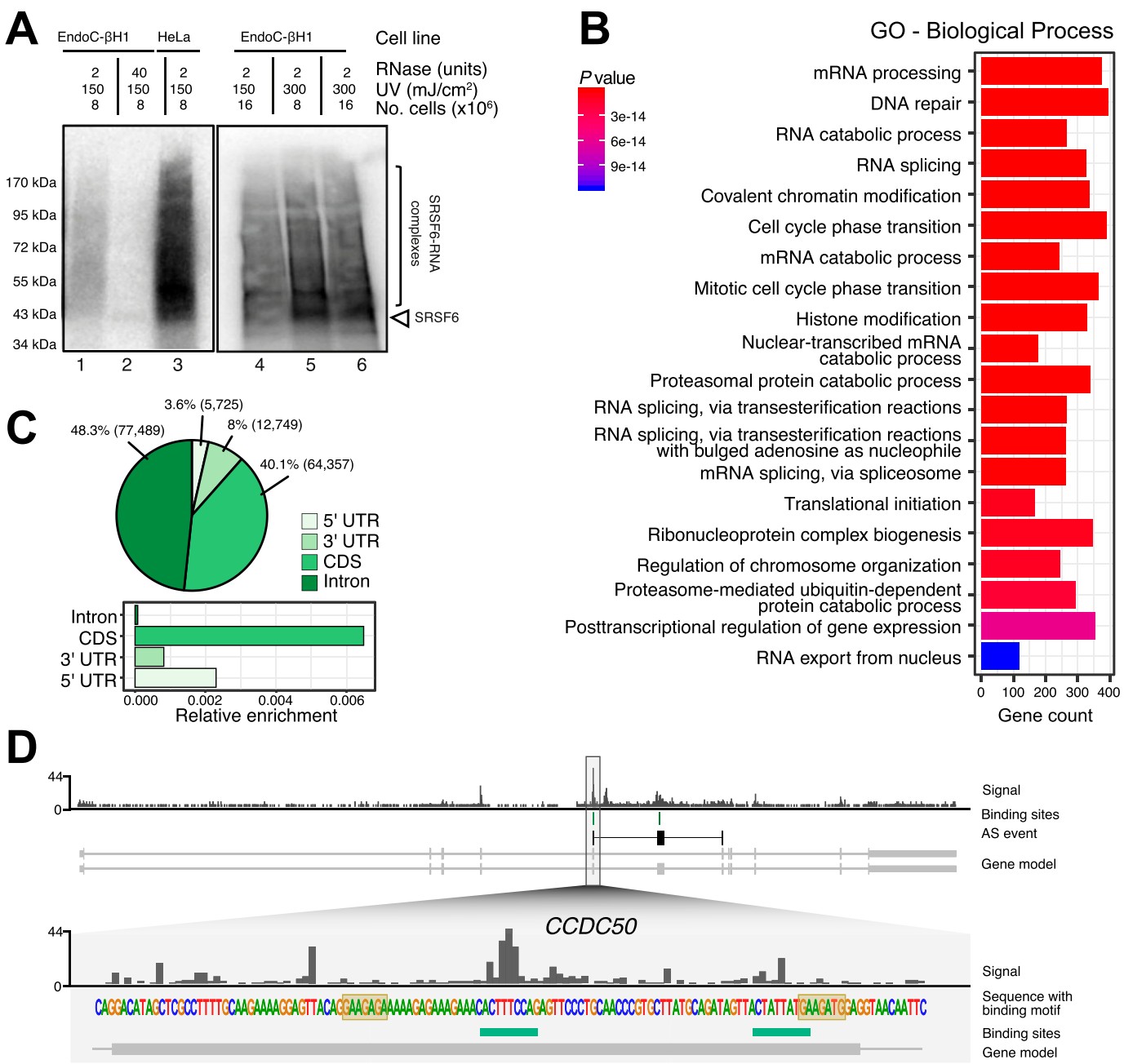

**Figure 1. iCLIP experiments show SRSF6 binding to thousands of transcripts in EndoC-βH1 cells.**
**(A)** Optimization of UV cross-linking conditions indicates that a combination of 300 mJ/cm² UV energy and 8 × 10⁶ cells gives highest yield. Autoradiographs of ³²P-labeled SRSF6–RNA complexes that immunoprecipitated from EndoC-βH1 (lanes 1, 2, 4–6) and HeLa cells (lane 3), separated by SDS–PAGE and immobilized on a nitrocellulose membrane. UV cross-linking was performed with different amounts of cells and using UV 254 nm with different irradiation energy, as indicated above. EndoC-βH1 cells were treated with low (2 U) and high (40 U) RNase concentration. High RNase concentration focuses the protein-RNA complexes to a defined band slightly above the expected molecular weight of SRSF6 (43 kD, arrowhead). **(B)** Gene Ontology enrichment in SRSF6-bound genes. Gene count refers to number of genes in the tested set that are associated with a given Gene Ontology term. *P*-value from hypergeometric distribution. **(C)** SRSF6 primarily binds coding sequences (CDS). The pie chart (top) shows distribution of SRSF6 binding sites per transcript region on protein-coding genes. Numbers indicate percentage or absolute number (brackets) of SRSF6 binding sites. The bar chart (bottom) shows relative enrichment per region, that is, number of binding sites normalized by summed length of respective bound transcript regions. **(D)** SRSF6 preferentially binds on exons of the *CCDC50* gene. Genome browser view of SRSF6 iCLIP data (signal of merged replicates), binding sites (green), and SRSF6 motif (yellow boxes). Selected transcript isoforms are shown below (GENCODE v29), with black boxes highlighting a SRSF6-regulated alternative exon together with the flanking constitutive exons. UTR, untranslated region.
Source data are available for this figure.

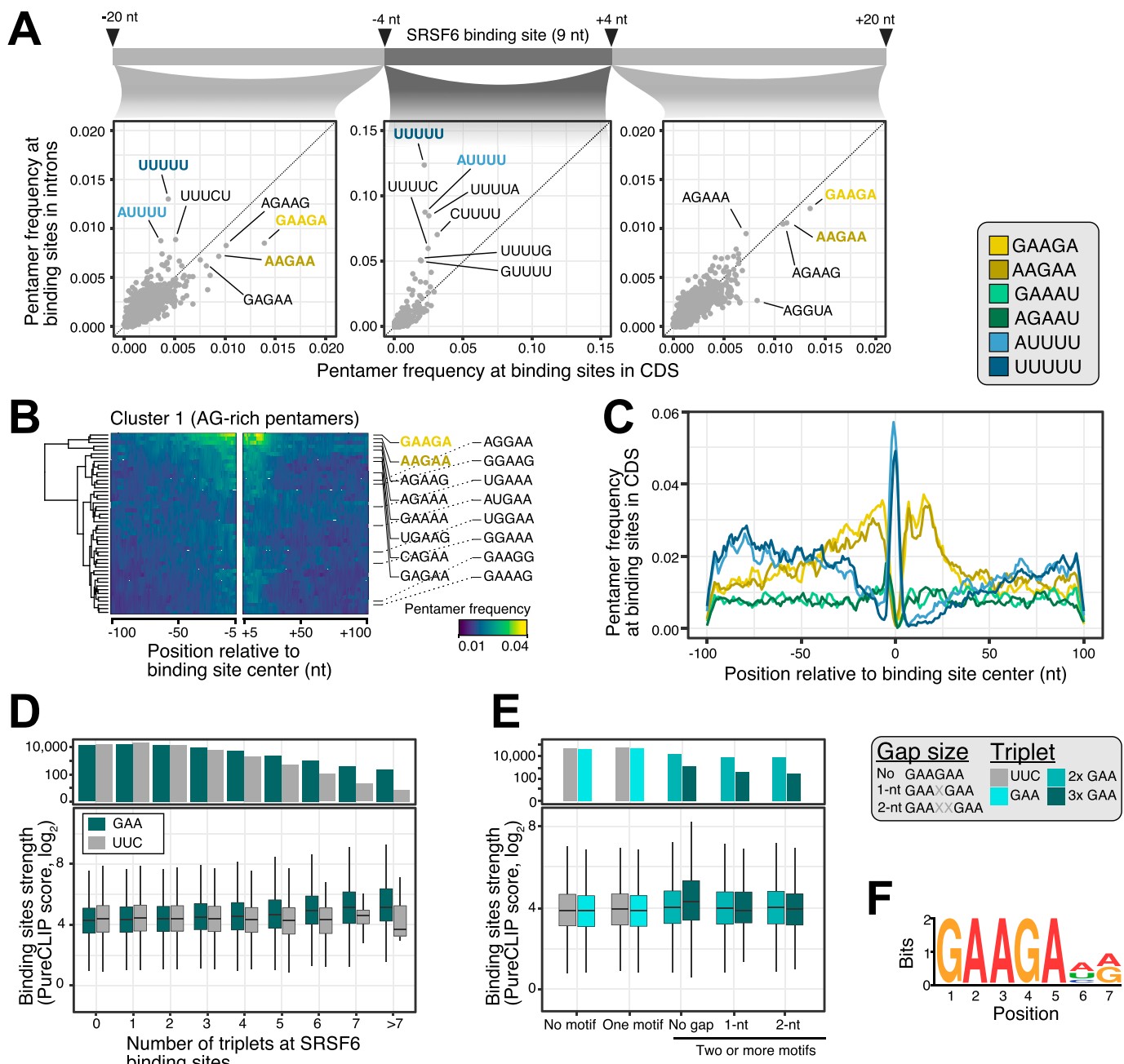

**Figure 2. SRSF6 specifically recognizes GAA motifs in exons.**
**(A)** SRSF6 binding sites in CDS frequently display AG-rich pentamers, contrasting the uridine-rich pentamers in introns, most prominently UUUUU. Scatter plots compare pentamer frequency within the 9-nt binding sites and in flanking 20-nt windows for SRSF6 binding sites in introns and CDS. Two most enriched pentamers of clusters 1 and 2 derived from hierarchical clustering of pentamer profiles are colored (see (B) and Fig S2A). **(B)** GAAGA and AAGAA enrich around SRSF6 binding sites in the CDS, whereas UUUUU marks binding site centers. Heat map shows cluster 1 from hierarchical clustering of pentamer profiles. Two most enriched pentamers are labeled and colored as in (A). Clusters 2 and 3 with uridine-rich and other pentamers are shown in Fig S2A and B. **(C)** SRSF6 positions towards end of motif-enriched stretches. Metaprofile shows pentamer frequencies in 201-nt window around SRSF6 binding sites. Two most enriched pentamers of three clusters from hierarchical clustering of pentamer profiles are shown. **(D)** Binding site strength increases with the number of GAA triplets. Boxplot (bottom) shows the distribution of binding site strengths ($log_2$-transformed PureCLIP score) for binding sites with a given number of GAA triplets within 30 nt from binding site center. Reverse complement UUC was used as control. Box represents quartiles, center line denotes 50th percentile, and whiskers extend to most extreme data points within 1.5× interquartile range. The bar chart (top) gives number of binding sites in each category. **(E)** Two or more GAA triplets in direct sequence are associated with increased binding site strength. Boxplot shows the distribution of binding site strengths ($log_2$-transformed PureCLIP score) for binding sites with no or one triplet (GAA or UCC) compared with two or more triplets in direct sequence or with 1-nt or 2-nt gaps. Visualization as in (D). **(F)** Motif enrichment analysis using DREME (Bailey, 2011) detected a purine-rich motif, reinforcing the role of GAA regions in SRSF6 binding. The motif is present at 25,148 SRSF6 binding sites.

indicating that SRSF6 precisely positions towards the end of the pentamer-enriched region (Fig 2C).

Common to most pentamers that clustered up- and downstream of the SRSF6 binding sites was the triplet GAA (Fig 2B). In support of a direct role of this triplet, a higher number of GAA triplets associated with stronger SRSF6 binding, starting from two up to seven or more GAAs (Fig 2D). A similar trend could not be observed for the reverse complement triplet UUC. Consistently, UUC triplets were less frequently observed in SRSF6 binding sites compared to GAA (Fig S2C). More precisely, at least two GAA triplets had to occur in direct succession to impart strong SRSF6 binding, whereas gaps of one or more nucleotides between the triplets were not tolerated (Fig 2E). This was also supported by a motif enrichment analysis using DREME (Bailey, 2011), which detected a GAA-rich consensus motif at 25,148 SRSF6 binding sites (19%; Figs 2F and S2D). Altogether, we conclude that SRSF6 preferentially binds to GAA-rich consensus motifs in CDS and positions at the end of motif-enriched regions.

## AS is coupled to SRSF6 positioning on cassette exons (CEs)

To evaluate the association between the SRSF6 binding and AS, we integrated the SRSF6 iCLIP with our published RNA sequencing (RNA-seq) data from EndoC-βH1 cells upon *SRSF6* KD (Juan-Mateu et al, 2018) which were presently re-analyzed as described in the Materials and Methods section. In agreement with our previous analysis (Juan-Mateu et al, 2018), we observed that *SRSF6* KD mainly resulted in CE skipping, with 975 and 237 CE showing significantly increased skipping and inclusion, respectively (false discovery rate < 0.05, absolute Δ percent spliced-in [|ΔPSI|] > 0.05) (Fig 3A and Table S1). The predominant directionality of the splicing changes pointed to the fact that SRSF6 generally drives exon inclusion (Jensen et al, 2014; Änkö, 2014). The splice sites of these exons were slightly weaker than at other exons in the same transcripts, indicating that splice site strength influences SRSF6 regulation (Fig S3A and B). Most AS events harbored one or more SRSF6 binding sites in the alternatively spliced region, indicating that they may include direct targets of SRSF6 regulation (Fig 3B).

In order to address the positioning of SRSF6, we generated metaprofiles of cross-link events on the up- and down-regulated exons ("RNA splicing map"). To avoid biases in the iCLIP signal that arise from differences in wild-type inclusion levels, we sampled two PSI-matched background sets of randomly selected exons with a comparable PSI distribution in wild-type EndoC-βH1 cells (Fig S3C–E). In line with SRSF6's role as a global splicing regulator, we observed substantial SRSF6 binding across all exons, extending into the immediately flanking intronic regions close to both splice sites (Fig 3C). Of note, this binding was significantly increased on the regulated exons, with distinct patterns on down- and up-regulated exons. Consistent with a direct enhancer function, exons with decreased inclusion upon *SRSF6* KD displayed more SRSF6 binding on the regulated exons (Figs 3C and S4A). Inversely, exons with increased inclusion featured massive binding in the flanking constitutive exons, suggesting that reinforcing the flanking exons can facilitate alternative exon skipping (Figs 3C and S4B). In support of this notion, up-regulation was predominantly seen for alternative exons, whereas down-regulation showed an almost equally

strong effect on exons that are constitutively spliced under normal conditions (Fig 3D).

In essence, our findings suggest that SRSF6 regulates AS in a position-dependent manner. As described for other SR proteins (Sanford et al, 2009; Han et al, 2011), binding within the alternative exon results in increased recognition, whereas strong binding to the flanking exons can reduce alternative exon inclusion.

## SRSF6-mediated AS reshapes the EndoC-βH1 cell transcriptome

We have previously shown that in parallel to AS, *SRSF6* KD impacted on gene expression (Juan-Mateu et al, 2018). A reanalysis of differentially expressed genes and a general overlay with AS changes upon *SRSF6* KD showed no evidence for a direct association between both types of regulatory events, for instance via nonsense-mediated mRNA decay (Fig S5A and B and Table S2). This indicated that rather than triggering global down-regulation, SRSF6-mediated AS may serve to specifically adapt transcripts and hence the encoded protein isoforms. Among the genes that were affected by differential expression, we detected the genes for many other splicing regulators (Fig S5C). These included *SRSF4* which was significantly up-regulated upon *SRSF6* KD, harbored a SRSF6-regulated AS event and showed strong SRSF6 binding along the complete transcript (Fig S5C and D). A cross-regulation between SR proteins and other splicing factors has been previously described (Pandit et al, 2013; Turunen et al, 2013; Brooks et al, 2015; Lareau & Brenner, 2015) and could affect the observed splicing effects due to partial compensation.

## SRSF6 regulates several susceptibility genes for type 1 and type 2 diabetes

We previously found that SRSF6 regulated AS of many genes involved in central β-cell functions, such as insulin secretion, evidencing its role as key splicing regulator for β-cells and thus suggesting a link to diabetes (Juan-Mateu et al, 2018). We, therefore, overlaid our data to a list of T1D and T2D susceptibility genes compiled from ImmunoBase (www.immunobase.org, accessed November, 2018), a genome-wide association studies (GWAS) catalog (https://www.ebi.ac.uk/gwas/, accessed November, 2018) and two recent publications (Wen & Yang, 2017; Udler et al, 2018). Intersection with our SRSF6 targets yielded five T1D susceptibility genes and 17 T2D susceptibility genes, including one shared between both sets, which harbored at least one significant CE with an SRSF6 binding site (out of 102 and 330, respectively; Fig 4A–D). Among the affected T1D susceptibility genes, *BCAR1*, *CENPO*, and *CDK2* displayed significant CE down-regulation upon *SRSF6* KD, whereas *LMO7* and *ITGB3BP* harbored one and two CEs, respectively, with increased inclusion (Fig 4A). At four of six affected exons, the associated SRSF6 binding sites harbored the GAA-rich consensus motif (Fig 4A), which we found to confer strong SRSF6 binding (Fig 2D and E).

Using semiquantitative RT-PCR, we tested the AS changes in independent experiments with EndoC-βH1 cells under control and *SRSF6* KD conditions. We successfully validated the splicing changes in seven out of nine (78%) selected CEs (Fig 4E). For the remaining two exons, in one case we could only amplify a single

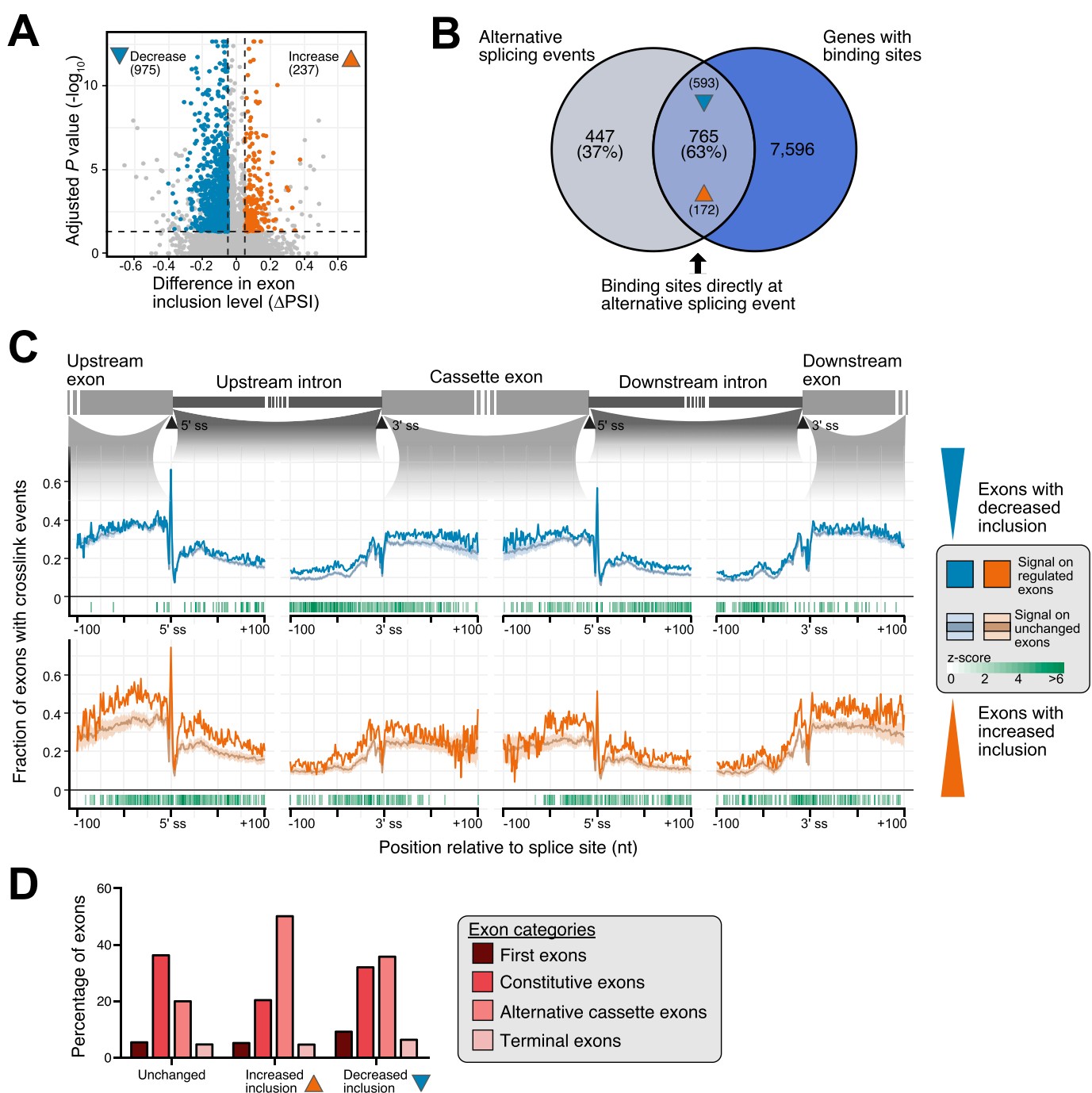

**Figure 3. SRSF6 binding on alternative and constitutive exons defines the splicing outcome in EndoC-βH1 cells.**
**(A)** *SRSF6* knockdown (KD) affects the inclusion of 1,212 cassette exons (CEs). Volcano plot shows $log_{10}$-transformed adjusted *P*-values against differences in "percent spliced-in" (ΔPSI) of CEs. Exons with significantly increased or decreased inclusion are colored (ΔPSI > 0.05, adjusted *P*-value < 0.05), according to reanalysis of our previously published RNA-seq data of EndoC-βH1 upon *SRSF6* KD (n = 5) (Juan-Mateu et al, 2018). **(B)** Most SRSF6-regulated CEs are directly associated with an SRSF6 binding site. Venn diagram depicts number and percentage of significantly regulated exons that harbor an SRSF6 binding site in the alternatively spliced region. Numbers in brackets at arrowheads specify exons with significantly increased and decreased inclusion in the overlap. **(C)** SRSF6 RNA splicing maps. SRSF6 shows more cross-link events on exons with decreased inclusion after *SRSF6* KD (blue, top), suggesting that SRSF6 binding is required for their inclusion, whereas exons with increased inclusion after *SRSF6* KD (orange, bottom) display more SRSF6 on the flanking constitutive exons. Metaprofile depicts fraction of exons with cross-link events at a given position within 100-nt on either side of indicated 3′ and 5′ splice sites. For comparison, mean and standard deviation are shown for a PSI-matched background distribution of unchanged exons (light blue and light orange lines, respectively; see Fig S3C–E), resulting from randomly sampled unchanged exons in EndoC-βH1 cells with comparable exon inclusion levels under control conditions. Only positive z-scores with adjusted *P*-value < 0.05 are shown. **(D)** Up-regulation upon *SRSF6* KD predominantly affects alternative exons, whereas down-regulation is almost equally observed for constitutive exons. The bar chart shows the contribution of different exon categories to exons with significantly increased or decreased inclusion upon *SRSF6* KD according to Exon Ontology analysis (Tranchevent et al, 2017). All remaining CEs (Unchanged) are shown as control.

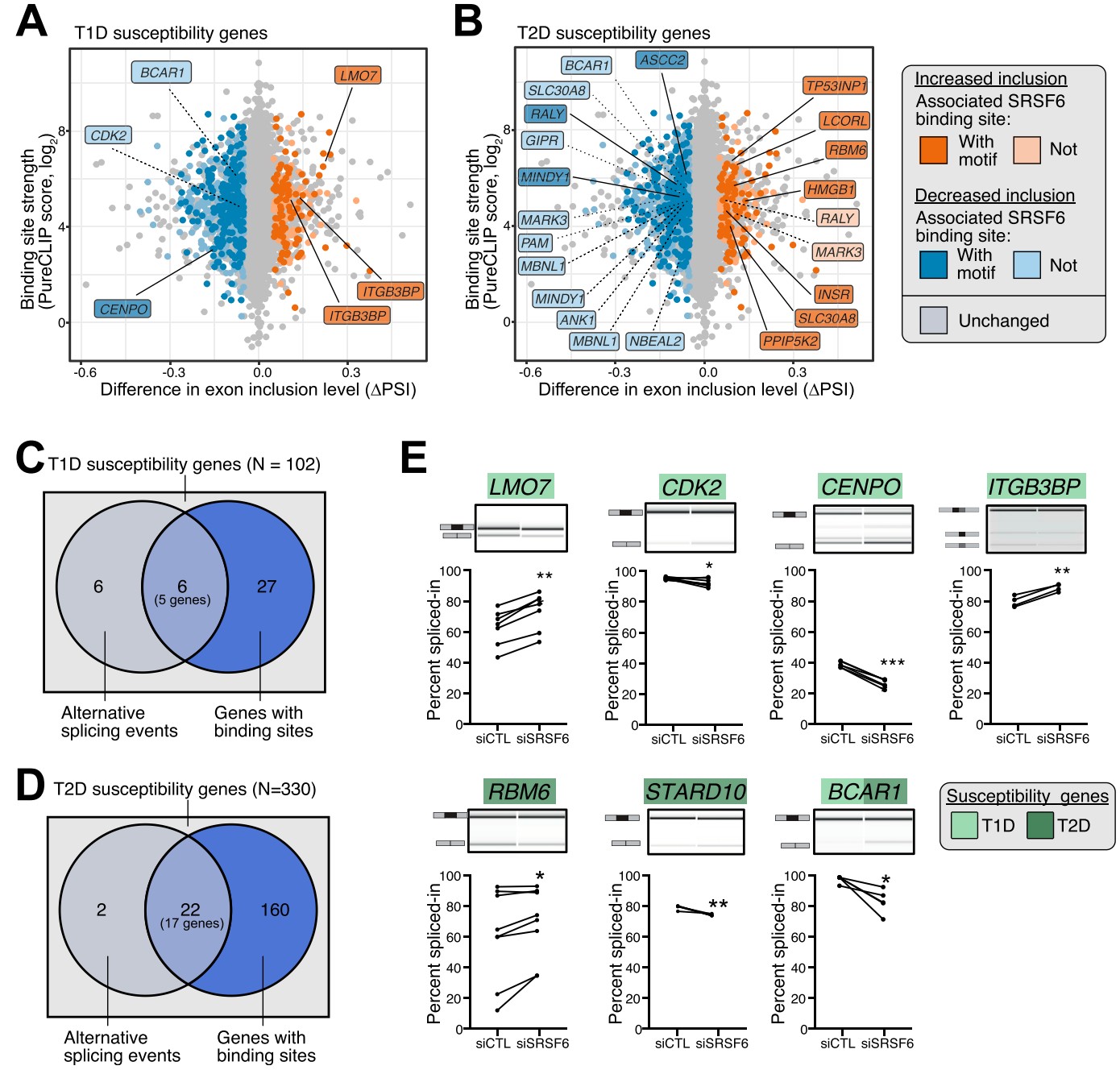

**Figure 4. Diabetes susceptibility genes are directly bound by SRSF6.**
**(A, B)** SRSF6 regulates alternative splicing events in several type 1 diabetes (T1D) (A) and type 2 diabetes (T2D) (B) susceptibility genes. Scatter plots show inclusion level differences (ΔPSI) against strength of strongest associated SRSF6 binding site (log$_2$-transformed PureCLIP score). Genes highlighted in dark blue and dark orange carry at least one SRSF6 binding site with a GAA-rich motif within 100 nt as shown in Fig 2F. Note that several genes harbor multiple SRSF6-regulated splicing events. **(C, D)** SRSF6 regulates 6 and 22 splicing events in 5 T1D (C) and 17 T2D (D) susceptibility genes, respectively. Venn diagrams depict overlap of SRSF6-regulated cassette exons and susceptibility genes. **(E)** Seven out of nine SRSF6-regulated exons in T1D (lightgreen) and T2D (darkgreen) susceptibility genes were validated by semiquantitative RT-PCR in control (siCTL) and SRSF6 KD (siSRSF6) EndoC-βH1 cells. Representative digital gel images (top) and quantifications with paired individual data points for 4–7 independent experiments (bottom) are shown for each splicing event. *P < 0.05, **P < 0.01, and ***P < 0.001, paired t test.
Source data are available for this figure.

band, precluding any AS affects, whereas in the other case, there was no significant difference between conditions (Fig S6A and B). The validated exons included all affected CEs in T1D susceptibility genes (*LMO7*, *ITGB3BP*, *CDK2*, and *CENPO*), two in T2D susceptibility genes (*RBM6* and *STARD10*), and one gene associated with both T1D and T2D (*BCAR1*) (Fig 4E). The splicing change affecting the T2D candidate gene *INSR* had already been validated in our previous study (Juan-Mateu et al, 2018).

Considering gene expression, further T1D and T2D susceptibility genes were affected in addition to the observed AS changes after *SRSF6* KD (Fig S7A and B). Several proteins encoded by the affected genes reside in a conjoint protein–protein interaction (PPI) network (Fig S7C). Via direct PPIs, they were further linked to additional susceptibility genes, namely *CDKN1B*, *IRS1*, *INS*, *TCF12*, and *CCND1*, suggesting a functional network of proteins encoded by susceptibility genes that could be affected by SRSF6.

Taking into account that SRSF6 was initially found as a downstream target of the diabetes candidate gene *GLIS3* in the rat cell line Ins1E (Nogueira et al, 2013), we tested for this association in human cells. Using RNA-seq data of pancreatic tissue samples from healthy individuals in the Genotype-Tissue Expression (GTEx) database (n = 127 pancreatic tissue samples, GTEx version 8) (https://gtexportal.org/home/, accessed on 09 May, 2020), we found a strong positive correlation between *GLIS3* and *SRSF6* gene expression (Fig 5A). Because an association with the splicing events could not be assessed in the GTEx data due to low signal for individual junctions, we depleted *GLIS3* in EndoC-βH1 cells using previously validated siRNAs (Nogueira et al, 2013) (Fig 5B). As previously described by us (Nogueira et al, 2013; Juan-Mateu et al, 2018), *GLIS3* depletion resulted in increased apoptosis, accompanied by a significant down-regulation of *SRSF6* mRNA expression (Fig 5C and D). Importantly, *GLIS3* depletion also triggered the expected splicing changes in the *LMO7* and *ITGB3BP* transcripts (Fig 5E). Three of the remaining genes (*CENPO*, *CDK2*, and *RBM6*) showed a trend in the expected direction, probably reflecting the milder *SRSF6* mRNA level reduction upon *GLIS3* depletion as compared

with the direct *SRSF6* KD (Fig S6C). For *STARD10*, there was a significant splicing modification but in the opposite direction from the one induced by *SRSF6* KD (Fig S6C). Finally, *BCAR1* showed only one band, precluding detectable splicing modification (Fig S6D). As a whole, these observations support, at least in part, the hypothesis that the splicing regulator SRSF6 acts downstream of the diabetes candidate gene *GLIS3*. GLIS3, however, has a broad impact on β-cell phenotype and survival (Taha et al, 2003; Senée et al, 2006; Dimitri et al, 2011; Nogueira et al, 2013), which most probably goes beyond its effects on SRSF6 regulation.

Altogether, we found that SRSF6 acts downstream of GLIS3 and affects the expression of multiple T1D and T2D susceptibility genes via AS changes and/or differential gene expression. The functional impact of these susceptibility genes deserves further molecular studies.

## Antisense oligonucleotides allow to reverse the *SRSF6* KD effect on *LMO7* splicing

Among the genes with SRSF6-regulated AS and direct SRSF6 binding, the T1D susceptibility gene *LMO7* was particularly interesting. *LMO7* is located on 13q22 and harbors an intronic SNP (rs539514) that is significantly associated with T1D (Bradfield et al, 2011). In our RNA-seq data, we found that inclusion of exon 10 (ENST00000341547) went up by more than 20% upon *SRSF6* KD in the EndoC-βH1 cells (Figs 4A and E and 6A). Importantly, the same splicing change could be triggered by treatment with the pro-inflammatory cytokines IL1β and IFNγ (Fig 6B), which mimic in vitro

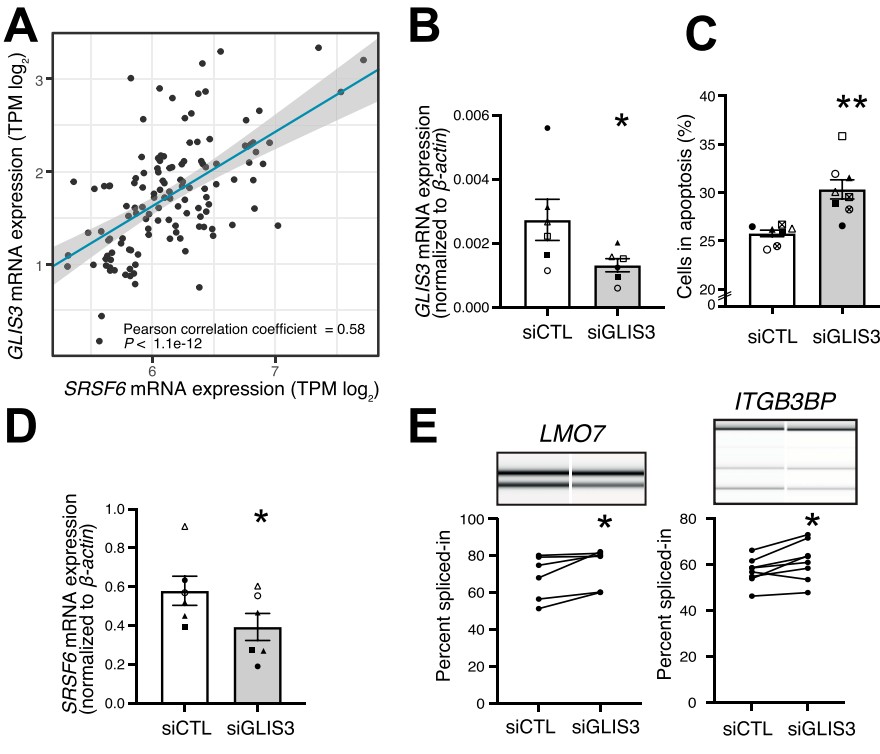

**Figure 5. SRSF6 acts downstream of GLIS3 and affects the splicing of diabetes susceptibility genes.**
**(A)** *SRSF6* gene expression correlates with *GLIS3* gene expression in human pancreatic tissue samples. Scatter plot compares gene expression (log₂-transformed transcripts per million, TPM) in pancreas samples of 127 healthy individuals from the Genotype-Tissue Expression (GTEx) database. Linear regression line (blue) with 95% confidence interval (gray corridor), Pearson correlation coefficient and associated *P*-value are indicated. **(B)** *GLIS3* was depleted in EndoC-βH1 cells by transfection with control (siCTL) or specific siRNA targeting *GLIS3* (siGLIS3) for 48 h. *GLIS3* mRNA expression was measured by quantitative real-time PCR (qRT-PCR) and normalized by the housekeeping gene *β-actin*. Mean ± SEM are shown for six independent experiments. **(C)** *GLIS3* depletion increases EndoC-βH1 cell apoptosis. Percentage of dead cells was counted after Hoechst and propidium iodine staining. Apoptosis was confirmed by microscopic evaluation of cell morphology and chromatin condensation. Mean ± SEM are shown for eight independent experiments. **(D)** *SRSF6* mRNA expression is decreased after *GLIS3* depletion. *SRSF6* mRNA expression was measured by qRT-PCR and normalized to *β-actin*. Mean ± SEM are shown for six independent experiments. **(E)** *GLIS3* depletion promotes alternative exon inclusion in the T1D susceptibility genes *LMO7* and *ITGB3BP*. Exon inclusion was quantified by semiquantitative RT-PCR and capillary gel electrophoresis. Representative digital gel images and quantification of paired individual data points for six independent

experiments are shown. *P < 0.05, **P < 0.01, and *** P < 0.001 against siCTL, paired *t* test. Source data are available for this figure.

some of the pro-inflammatory conditions observed in islets of Langerhans from T1D patients (Eizirik et al, 2020). This coincided with an up-regulation in *LMO7* expression seen in published RNA-seq data (Ramos-Rodríguez et al, 2019), which we similarly detected after *SRSF6* KD (Figs 6C and S7A). Mirroring the global SRSF6 regulatory profile (Fig 3C), the *LMO7* transcript displayed strong SRSF6 binding on the flanking exons, most prominent on the upstream constitutive exon 9 (Fig 6A). This indicated that SRSF6 kept down the inclusion of exon 10 under normal conditions by reinforcing the flanking exons. The repression was then lost upon *SRSF6* KD.

To test whether AS can be modulated in EndoC-βH1 cells, we explored the application of antisense oligonucleotides (ASOs). The aim was to shift the splicing of *LMO7* exon 10 towards exon skipping, thereby restoring its splicing pattern to the basal levels. To achieve this, we designed two ASOs that blocked the 3′ and 5′ splice sites of *LMO7* exon 10 (ASO-5ss and ASO-3ss, respectively; Fig 6D and Table S3). An ASO sequence that had no homology in the human genome was used as a negative control (ASO-Ctrl) (Ziegler et al, 1997). All designed ASOs contained 2′-*O*-methyl RNA nucleosides and a full-length phosphorothioate backbone to improve their uptake and half-life, by increasing their intracellular stability against enzymatic degradation (Khvorova & Watts, 2017). The ASOs were transfected into EndoC-βH1 cells at three different concentrations (10, 50 and 200 μM; Fig 6E). Importantly, ASO treatment triggered *LMO7* exon 10 skipping, and increasing the ASO concentration, particularly of ASO-5ss, augmented this effect, reaching a decrease in inclusion of 30–40% at 200 μM. A similar trend was observed for ASO-3ss, although it did not reach significance. To test whether the effect persisted in presence of *SRSF6* KD, we co-transfected the ASOs with siCTL or siSRSF6. Whereas *SRSF6* KD moderately increased exon inclusion, the co-transfection with ASOs significantly potentiated *LMO* exon 10 skipping (Fig 6F). This underlined that the designed ASOs were able to bind pre-mRNA sequences before splicing, thus preventing the recognition and thereby promoting the removal of exon 10 from the mature mRNA molecule. Together, these results indicate that ASOs can be applied in human β-cells to modulate AS, thereby offering means to counteract the dysregulation of splicing factors induced by pro-inflammatory cytokines.

## Discussion

*GLIS3*, a susceptibility gene for T1D and T2D and monogenic forms of the disease (Taha et al, 2003; Senée et al, 2006; Dimitri et al, 2011), decreases the expression of the splicing regulator SRSF6, and we have previously shown that SRSF6 inhibition hampers human β-cell function and viability (Juan-Mateu et al, 2018).

Here, we investigated the mechanisms of splicing regulation by the splicing factor SRSF6 in the human pancreatic β-cell line EndoC-βH1 and its impact on diabetes susceptibility genes. Using iCLIP, we identified more than 185,000 SRSF6 binding sites in nearly 9,000 genes from EndoC-βH1 cells. As previously described (Jensen et al, 2014; Müller-McNicoll et al, 2016), and in line with its splice enhancer function, SRSF6 showed a strong preference for exons,

predominantly in the CDS. In many cases, binding extended beyond the exon-intron boundaries, additionally covering the splice site regions. Similarly, other SR proteins modulate splicing by binding to an intronic splicing enhancer (Lou et al, 1998), or branch-point sequences (Shen et al, 2004; Cho et al, 2011a; Änkö et al, 2012). Finally, the observed secondary binding to 5′ and 3′ UTRs may reflect non-splicing-related functions of SRSF6, such as regulation of mRNA stability, export or translation, as it has been observed for other SR proteins (Lemaire et al, 2002; Kim et al, 2014; Müller-McNicoll et al, 2016).

Based on previous analysis (Tacke & Manley, 1995; Liu et al, 1998; Sanford et al, 2008; Änkö et al, 2012; Bradley et al, 2015), SRSF6 and other SR proteins recognize short, 4–8 nt long, degenerate sequences. In the present study, we substantially refined the current knowledge about SRSF6's RNA binding specificity. Using in-depth sequence analyses, we found that SRSF6 binds to repetitions of two or more GAA triplets which should not be interrupted by other nucleotides. The motif enrichment around binding sites extended up to 100 nt and binding strength progressively grew with increasing number of triplets, suggesting that multiple SRSF6 proteins may assemble on a given exon to reach effective splicing enhancement. Alternatively, the repetition of motifs could favor the binding of additional SR proteins or other RBPs recognizing the same motifs. The second scenario may confer selective advantage for the recognition of degenerative motifs to ensure exon inclusion, also by offering the possibility for compensatory regulatory mechanisms among SR proteins (Pandit et al, 2013).

Of note, the SRSF6 consensus motif found in our study differs from previously reported motifs that were obtained in other cell types using SELEX (systematic evolution of ligands by experimental enrichment) and RNA immunoprecipitation experiments (Screaton et al, 1995; Liu et al, 1998; Änkö, 2014; Park et al, 2019). It does, however, precisely fit to the consensus motif that was more recently derived from SRSF6 iCLIP experiments in mouse cells (Müller-McNicoll et al, 2016). Moreover, our GAA-rich consensus motif is more similar to sequences previously shown to function as binding sites for SRSF1, SRSF4, and SRSF7 (Zheng et al, 1997; Sanford et al, 2009; Änkö et al, 2010, 2012). Together, these observations suggest that the RNA sequence specificity of SRSF6 in vivo is considerably different from what was previously reported based on in vitro SELEX experiments. Importantly, SR protein motifs are commonly used for the in silico prediction of exonic splicing enhancers which is used for mechanistic studies but also for the characterization of SNPs or somatic variants in human diseases (Cartegni et al, 2003; Solovyev & Shahmuradov, 2003; Desmet et al, 2009). Our findings therefore have important implications for the interpretation of such variants and their putative role on splicing regulation.

We previously reported that SRSF6 regulates a network of AS events in pancreatic β-cells which impact on key processes like β-cell survival and insulin secretion (Juan-Mateu et al, 2018). Importantly, by integrating the re-analyzed data with the present iCLIP data we demonstrate that most of the observed AS events are associated with direct binding by SRSF6. Moreover, as previously described for other splicing factors (Ke & Chasin, 2011; Pandit et al, 2013; Bradley et al, 2015), we show that SRSF6 exhibits a context- and position-dependent mode of splicing regulation. We carefully controlled this analysis for abundance differences of the exons,

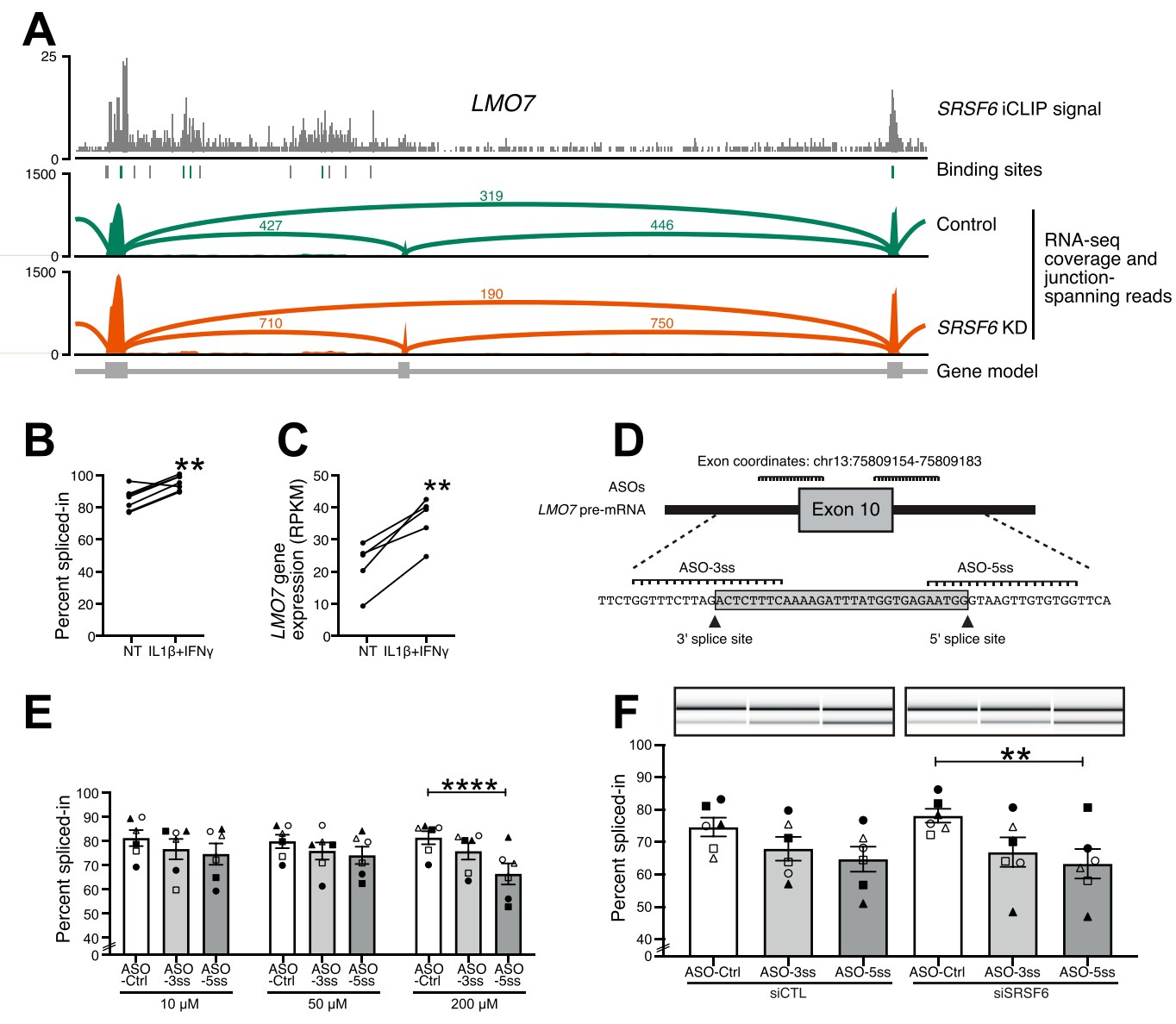

**Figure 6. Antisense oligonucleotides promote *LMO7* exon 10 skipping in EndoC-βH1 cells.**
**(A)** SRSF6 binds *LMO7* and suppresses inclusion of exon 10. Genome browser view depicts SRSF6 iCLIP data (merged replicates) and binding sites (green, with motif; gray, without motif) together with RNA-seq read coverage from control and *SRSF6* KD EndoC-βH1 cells. Lines indicate exon-exon junctions with numbers of supporting reads. In line with the global RNA splicing map for SRSF6 (Fig 3C), SRSF6 binding in the preceding constitutive exon down-regulates *LMO7* exon 10 inclusion. **(B)** Pro-inflammatory conditions increase *LMO7* exon 10 inclusion. Quantification of semiquantitative RT-PCR for *LMO7* exon 10 inclusion under basal conditions (NT, not treated) and after exposure to pro-inflammatory cytokines (IL1β + IFNγ) for 48 h. Paired individual data points are shown for six independent experiments. **P-value < 0.01, paired *t* test. **(C)** *LMO7* gene expression goes up in EndoC-βH1 cells exposed to IL1β + IFNγ. Analysis of *LMO7* gene expression (RPKM, reads per kilobase per million mapped reads) in published RNA-seq data (n = 5) (Ramos-Rodríguez et al, 2019). **P-value < 0.01, paired *t* test. **(D)** Antisense oligonucleotides (ASOs) were designed to target the 3′ and 5′ splice site of *LMO7* exon 10 (ASO-3ss and ASO-5ss, respectively). Schematic representation of the pre-mRNA sequence of *LMO7* exon 10 and flanking intronic regions. Annealing of the ASOs to the indicated positions is predicted to interfere with splice site recognition and thereby to reduce *LMO7* exon 10 inclusion. **(E)** Targeting the splice sites decreases inclusion of *LMO7* exon 10, most prominently with ASO-5ss at 200 µM. The bar chart depicts quantification of *LMO7* exon 10 inclusion after transfection with different ASO concentrations (10, 50, and 200 µM) for 48 h. Mean ± SEM and individual data points are shown for semiquantitative RT-PCR measurements (n = 6). ANOVA followed by Bonferroni correction was used to compare three ASOs at each concentration. ****P-value < 0.0001. **(F)** ASOs promote *LMO7* exon 10 skipping in presence of *SRSF6* KD. Representative digital gel images (top) and quantification (bottom) of semiquantitative RT-PCR in ASO-treated EndoC-βH1 cells under control conditions (siCTL) and after *SRSF6* KD (siSRSF6), co-transfected for 48 h with 50 µM of ASO-Ctrl, ASO-5ss, or ASO-3ss. Results are mean ± SEM of six independent experiments. ANOVA followed by Bonferroni correction was used to compare three ASOs in control or after *SRSF6* KD. **P-value < 0.01.
Source data are available for this figure.

ensuring that our observations reflect genuine regulatory RNA binding profiles rather than intrinsic biases which commonly impair a direct comparison of up- and down-regulated exons (Fig S3C–E). We observed increased SRSF6 binding within the exons that

are down-regulated upon *SRSF6* KD, supporting that the binding of SRSF6 mainly contributes to exon inclusion. This is in line with early in vitro studies reporting that SR proteins bind pre-mRNAs mainly at intronic splicing enhancers and exonic splicing enhancers and

recruit spliceosomal components to the neighboring splice sites (Graveley et al, 1999). To date, the precise mechanism by which SR proteins favor splice site recognition remains debatable. It has been described that SRSF1 assists the recruitment of the U1 small nuclear ribonucleoprotein (snRNP) component U1-70K, thus facilitating donor site recognition (Wu & Maniatis, 1993; Cho et al, 2011a). Moreover, similar to other SR proteins (Fu & Maniatis, 1992), SRSF6 binding at neighboring splice sites may potentiate the recruitment of U1 snRNP and U2 auxiliary factor 2 (U2AF) to the 5′ and 3′ splice sites, respectively, of the regulated exons (Long & Cáceres, 2009). Alternatively, SRSF6 binding may interfere with the binding of negative regulators, such as heterogeneous nuclear ribonucleoproteins (hnRNP) (Cáceres et al, 1994; Zhu et al, 2001; Long & Cáceres, 2009). Although some of the down-regulated exons had weaker splice sites compared with neighboring exons, we also observed a considerable number of constitutive exons, suggesting that SRSF6 binding can equally reinforce the splicing of constitutive and alternative exons (Shen & Green, 2006; Long & Cáceres, 2009). Surprisingly, more than 200 exons went up upon SRSF6 KD, suggesting that their inclusion is normally attenuated by SRSF6. These up-regulated exons displayed strong SRSF6 binding on the flanking constitutive exons and were themselves enriched for alternative exons with weaker splice sites. Together, these observations suggested that SRSF6 can hamper the inclusion of weak exons by reinforcing the neighboring exons. A similar scenario has been described for other splicing-regulatory proteins (Sanford et al, 2009; Han et al, 2011). A possible mechanism would be an increased splicing efficiency or faster splicing of the neighboring exons, which has been previously shown to repress alternative exons (Eperon et al, 1993; Sanford et al, 2009; Han et al, 2011). A prime example for this regulatory RNA binding profile is LMO7 exon 10 (see below).

We observed that SRSF6 KD influences the expression of many other splicing regulators, including other SR and hnRNP proteins. Similar to, for example, SRSF1 (Sanford et al, 2009), this effect on splicing factors and other RNA-related pathways highlights the potential role of SR proteins in maintaining RBP expression homeostasis, possibly through post-transcriptional regulatory mechanism such as AS coupled with nonsense-mediated mRNA decay (Lareau et al, 2007; Ni et al, 2007; Barberan-Soler & Zahler, 2008; Saltzman et al, 2008). As a whole, the present and previous findings suggest an interplay between SR proteins and other splicing regulators to ultimately determine the splicing outcome.

In the pancreatic β-cells, SRSF6 expression is regulated by the diabetes susceptibility gene GLIS3 (Nogueira et al, 2013), which encodes an important transcription factor for β-cell development and maintenance of the differentiated phenotype (Kang et al, 2009; Wen & Yang, 2017; Scoville et al, 2020). SNPs that lower GLIS3 expression increase the risk of both T1D and T2D (Barrett et al, 2009; Dupuis et al, 2010; Cho et al, 2011b; Steck et al, 2014; Winkler et al, 2014), whereas inactivating mutations in the gene itself cause severe neonatal diabetes (Taha et al, 2003; Senée et al, 2006; Dimitri et al, 2011). In contrast to many other putative T1D risk loci, the GLIS3-associated risk is not shared with other autoimmune diseases, emphasizing its effects at the β-cell level (Redondo & Concannon, 2020), probably by augmenting β-cell susceptibility to immune and metabolic stressors (Nogueira et al, 2013; Liston et al, 2017). Interestingly, the pro-apoptotic impact of GLIS3 inhibition in pancreatic β-cells is mediated at least in part via down-regulation of SRSF6 and consequent changes on transcripts that exert a crucial role on β-cell function and survival (Nogueira et al, 2013;

Juan-Mateu et al, 2018). These observations suggest the possibility of a crosstalk between diabetes susceptibility genes and splicing factors. To test this hypothesis, we specifically looked for SRSF6-regulated splicing events with associated binding sites in a compiled list of more than 400 T1D and T2D susceptibility genes. We found that SRSF6 KD significantly affected 28 AS events in well-documented diabetes susceptibility genes (five T1D susceptibility genes, and 17 T2D susceptibility genes) (Barrett et al, 2009; Bradfield et al, 2011; Pociot, 2017). Although still deserving detailed molecular analyses about their functional effect in β-cells, these susceptibility genes are generally predicted to mediated gene–environment interactions in diabetes. Our results thereby suggest that SRSF6 regulation can influence the architecture of the disease.

Splicing-modulating molecules such as ASOs have been developed to remodel disease-causing AS switches, with some of these molecules being in advanced stages of clinical trial or even in clinical practice (Spitali & Aartsma-Rus, 2012; Havens & Hastings, 2016; Yin & Rogge, 2019). Structurally, ASOs are single-stranded oligonucleotides of 15–25-nt length, that are usually chemically modified to facilitate their intracellular stability. However, despite their widespread usage in other cell models, pancreatic β-cells have been described as particularly refractory to ASO internalization (Hung et al, 2013). As a proof of concept for the potential of ASOs in pancreatic β-cells, we focused on the T1D candidate gene LMO7 because it harbors an intronic SNP (rs539514) that is highly associated with T1D (Bradfield et al, 2011). Whereas the role of LMO7 in pancreatic β-cells remains unknown, in other cell types it is related to PPIs at focal adhesion and adherens junctions, regulation of actin dynamics, and transcriptional regulation (Ooshio et al, 2004; Holaska et al, 2006; Hu et al, 2011). Notably, we found that SRSF6 KD up-regulated the inclusion of exon 10 in the LMO7 gene. Therefore, ASOs targeting the 3′ and 5′ splice sites may hamper the binding of spliceosomal components, thus preventing exon recognition and promoting exon exclusion. Importantly, we demonstrate that ASOs targeting the 5′ splice site significantly reduced inclusion of LMO7 exon 10. Together, our results suggest that splicing modulation by ASOs may be a promising approach for the future modulation of genes with potential impact on β-cell survival and function.

In summary, we identified the RNA binding profiles of SRSF6 in the human pancreatic β-cell line EndoC-βH1. Moreover, we showed that SRSF6, which is regulated by the diabetes candidate gene GLIS3, modulates the splicing of other diabetes susceptibility genes, suggesting the presence of an AS-regulated network with a putative impact on diabetes risk. The functional impact of the observed splicing modification will require future functional validation. We also demonstrated the possibility of modulating these events in EndoC-βH1 cells by specific ASOs. It will be of interest to investigate whether similar AS-related networks are present in other immune-mediated and/or genetically determined diseases.

# Materials and Methods

### Culture of human cells, gene silencing, and cytokine treatment

The human pancreatic β-cell line EndoC-βH1 was kindly provided by Dr R Scharfmann (Institut Cochin, Université Paris Descartes,

Paris, France) (Ravassard et al, 2011). EndoC-βH1 cells were cultured in DMEM containing 5.6 mmol/l glucose (Gibco, Thermo Fisher Scientific), 2% BSA fraction V, fatty acid free (Roche), 50 μmol/l 2-mercaptoethanol (Sigma-Aldrich), 10 mmol/l nicotinamide (Calbiochem), 5.5 μg/ml transferrin (Sigma-Aldrich), 6.7 ng/ml selenite (Sigma-Aldrich), 100 U/ml penicillin + 100 μg/ml strepto-mycin (Lonza) in matrigel–fibronectin–coated plates (Brozzi et al, 2015). HeLa cells were cultured in DMEM with 4.5 g/l D-glucose, enriched with 10% FBS and 100 U/ml penicillin + 100 μg/ml strep-tomycin (Lonza).

EndoC-βH1 were transfected with siRNA targeting *SRSF6* as described previously (Juan-Mateu et al, 2018); Allstars Negative Control siRNA (QIAGEN) was used as a negative control (siCTL). The siRNA sequences are listed in Table S4.

To test the effect of pro-inflammatory cytokines, EndoC-βH1 were exposed to the human cytokines interleukin-1β (50 U/ml; R&D Systems) and interferon-γ (1,000 U/ml; Peprotech) for 48 h, as previously described (Eizirik et al, 2012).

## Cell viability

The percentage of viable EndoC-βH1 cells was assessed using fluorescence microscopy after 20 min incubation with the DNA-binding dyes Hoechst 33342 (5 mg/ml; Sigma-Aldrich) and propi-dium iodide (5 mg/ml; Sigma-Aldrich), as described (Kutlu et al, 2003). Viability was evaluated by two independent observers, one of them unaware of sample identity. Results are expressed as per-centage of apoptosis, calculated as number of apoptotic cells/total number of cells.

## Western blot

Total protein extracts were obtained from EndoC-βH1 using Laemmli buffer, and separated on 8% SDS–PAGE. The nitrocellulose membranes were probed using specific primary antibodies, anti-SRSF6 (Anti-SRSF6/SRP55 [aa250-300] LS-C290327; LifeSpan Bio-science) diluted 1:1,000, anti-α-Tubulin (T5168; Sigma-Aldrich) diluted 1:5,000, in 1× TBST (Tris-buffered saline, 0.1% Tween 20) with 5% nonfat dry milk. After overnight incubation with the primary antibodies at 4°C, the membranes were probed for 1 h at room temperature with peroxidase-conjugated secondary antibodies (listed in Table S5). Detection of immunoreactive bands was per-formed using chemiluminescent substrate (SuperSignal West Femto, Thermo Fisher Scientific) using a Bio-Rad chemi DocTM XRS+ system (Bio-Rad Laboratories). The densitometric values were obtained by ImageLab software (Bio-Rad Laboratories) quantifi-cation and corrected against α-Tubulin as loading control, after background subtraction.

## iCLIP library preparation

iCLIP libraries were prepared based on a previously described protocol (Huppertz et al, 2014; Sutandy et al, 2016). Because of low replication rate of EndoC-βH1 cells, iCLIP cross-link conditions were first optimized in HeLa cells, and HeLa cells were then used as positive control (Figs 1A and S1C). The iCLIP libraries were prepared from four replicates of ~8 × 10$^6$ EndoC-βH1 (Fig S1D). The cells were

UV-irradiated with 254 nm UV with 300 mJ/cm$^2$ to induce cross-linking between SRSF6 and the interacting RNAs (König et al, 2010). Partial RNase digestion was performed by adding 2 U of RNase I (Ambion) to the sample lysates. The SRSF6–RNA complexes were immunoprecipitated with a specific anti-SRSF6 antibody (Anti-SRSF6/SRP55 [aa250-300] LS-C290327, LifeSpan Bioscience). The prepared iCLIP libraries were sequenced as 75-nt single-end reads on an Illumina HiSeq 2500 sequencing system. Sample barcodes used in this experiment are listed in Table S6.

## iCLIP data processing

Initial quality checks were applied to all reads before and after quality filtering using FastQC (version 0.11.5, available online at: https://www.bioinformatics.babraham.ac.uk/projects/fastqc/) (Andrews, 2010). Reads were filtered based on the sequencing qualities (Phred score) of the barcode region. Reads with more than one position with a sequencing quality <20 in the sample barcode (position 4–7) or with any position having a sequencing quality <17 in the random barcode regions (positions 1–3 and 8–9) were removed. Remaining reads were kept for further analysis and de-multiplexed based on the sample barcode (positions 4–7 of the reads) with no allowed mismatches using Flexbar (version 3.0.0) (Dodt et al, 2012). Sample barcodes are available in Table S6. Remaining adapter se-quences (AGATCGGAAGAGCGGTTCAG) were trimmed from the right end of the reads using Flexbar (Dodt et al, 2012), requiring an overlap of read and adapter of at least 1-nt and allowing one mismatch in 10-nt. Barcode regions (first 9-nt) were trimmed off at the left end of the reads and added to the read names, such as that barcode information was kept for downstream analysis. Reads shorter than 15-nt after adapter and barcode trimming were discarded. Trimmed reads were then mapped to the human genome (assembly version GRCh38) (Frankish et al, 2019) using STAR (version 2.5.2b) (Dobin et al, 2013) with two mismatches allowed and turned off soft-clipping on the 5′ end (the bases in the 5′ of the read were part of the alignment). Only uniquely mapped reads were kept for further analysis. Duplicate reads were marked using the *dedup* function of the bamUtils tool suit (http://github.com/statgen/bamUtil). Marked duplicates with identical random barcodes were removed as they were considered technical duplicates, whereas biological dupli-cates showing distinct random barcodes were kept for further analysis.

## Binding site definition

Processed read counts from all four replicates were merged into a single file for each strand and subjected to a peak calling step using PureCLIP (version 1.0.0) (Krakau et al, 2017) with default parameters. The resulting single nucleotide-wide significant cross-link sites were filtered to remove the sites with the 5% lowest PureCLIP scores. All remaining sites were merged into 9-nt-wide non-overlapping binding regions as described in Busch et al, 2019.

In brief, this was performed by clustering all significant cross-link sites closer than 8-nt into regions. Resulting regions that were shorter than 3-nt were removed. Equal-sized binding sites were achieved by either extending regions shorter than 9-nt, or by it-eratively splitting up regions larger than 9-nt. In both cases, the binding site center was defined by the position with the highest number of cross-link events (peak summit). We also required each

binding site to contain at least two positions covered with significant cross-link sites from the initial PureCLIP analysis. Next, reproducibility was established by requiring support of at least three of four replicates for each binding site, meaning that the binding site harbored more cross-link events from the replicate-specific threshold. This threshold was set to the 20% quantile of the distribution of cross-link events in all binding sites for the given replicate. This procedure yielded a total of 185,266 binding sites.

## Genomic distribution of SRSF6 binding sites

Gene and transcript annotations were obtained from GENCODE (release 29), and filtered for gene support (≤2) and transcript support level (≤3). Binding sites were overlaid with the filtered annotations, and overlapping annotations were removed. Of note, the latter filter excluded most SRSF6 binding sites in the T2D susceptibility gene *STARD10*, including those in the region of the regulated alternative exon. For all following analysis, only binding sites overlapping a single protein-coding gene were retained. On transcript level, a binding site was assigned to one of the four regions: intron, CDS, 3' UTR, or 5' UTR. Overlaps with multiple different transcript regions were resolved by applying a majority vote followed by a hierarchical rule, selecting intron > CDS > 3' UTR > 5' UTR. For Fig 1C, the number of binding sites was normalized by the summed lengths of the respective bound transcript regions. Following this procedure, a total of 160,320 binding sites in 8,533 genes could be assigned to a unique transcription region and they were used for all subsequent analyses.

## Motif definition

The local sequence content around SRSF6 binding sites was assessed by counting pentamer frequencies for 5,000 randomly sampled SRSF6 binding sites from each transcript region considered. Counts were obtained for the 9-nt binding site itself, as well as the 20-nt flanking regions up- and downstream. In Fig 2A, we compared the mean count of each pentamer for each window between binding sites in introns and CDS. For positional profiles (Fig 2C), overlapping pentamers were count on all nucleotides in the windows and divided by total number of considered binding sites. Heat maps in Figs 2B and S2A and B were obtained from $k$-means clustering with three centroids of frequency profiles for top 300 pentamers with highest frequency in SRSF6 binding sites in the CDS. To dissect the binding sequence further, the number of GAA versus UUC triplets was counted in the 49-nt-wide windows and compared with the respective PureCLIP score (Krakau et al, 2017). All counting operations were performed with the Biostrings package in R (Pagès, 2020).

The de novo motif search was performed using DREME (version 5.1.1, http://meme-suite.org/tools/dreme) (Bailey, 2011) on 201-nt-long sequence windows on SRSF6 binding sites in all transcript regions as input. The motif in Fig 2F corresponds to the second hit, preceded by a U-rich motif. For Fig S2D, the position weight matrix of the DREME motif in Fig 2F was taken as input to search again in all 201-nt windows using FIMO (Grant et al, 2011).

## RNA sequencing and data processing

The five independent RNA sequencing (RNA-seq) experiments of EndoC-βH1 cells exposed to control (siCTL) or *SRSF6* KD (siSRSF6)

used in the present study were previously published by our group (Juan-Mateu et al, 2018) (available at Gene Expression Omnibus [GEO] under accession number GSE98485). For the present study, the five paired RNA-seq replicates were re-analyzed as described herein. Initial sequence quality of the reads was monitored with FastQC (version 0.11.5) (Andrews, 2010). Adapter sequences were removed from all 3' ends and all resulting reads were subjected to a window-based quality trimming using Flexbar (Roehr et al, 2017) with default parameters (version 2.17.2). Reads were mapped with STAR (version 2.5.3a) (Dobin et al, 2013) to the human genome (GRCh38) with GENCODE gene annotations release 29 (Frankish et al, 2019). Thereby soft-clipping was enabled, but no multimapping reads were allowed with a maximum of two mismatches. All downstream analyses were performed using the human genome version GRCh38 with GENCODE gene annotations release 29.

## Differential expression analysis

Differential expression analysis was performed in R using DESeq2 (version 1.22.2) (Love et al, 2014). Mapped reads were counted in annotated exons using genomicAlignments (version 1.18.1) in "union" mode. The resulting count matrix was filtered for genes with at least 10 reads and the DESeq2 model was fitted with the formula "design ≅ pair + condition," to account for the paired arrangement of control and KD samples. Genes were considered significant if passing below an adjusted $P$-value < 0.001 (Benjamini-Hochberg correction), yielding a total of 4,126 differentially expressed genes. An additional filter on the absolute $\log_2$-transformed fold-change ($|LFC| > 1$) resulted in 106 differentially expressed genes. Genes encoding for splicing regulators were identified based on association with the Gene Ontology term GO: 0008380 "RNA splicing" (Fig S5C).

## AS analysis

AS analysis was performed using rMATS-turbo (version 4.0.1) (Shen et al, 2014) with default parameters, only specifying read type, read length and strand specificity (-t paired, –readLength 98, and –libType fr-unstranded). Because rMATS requires all reads to have the same length, we re-processed all reads to a fixed length of 98-nt after removing adapter sequences using Flexbar (Dodt et al, 2012). Mapping was again performed with STAR (Dobin et al, 2013) using the same settings as described above, but without soft-clipping. Based on our previous study (Juan-Mateu et al, 2018), we determined that CE events represent >50% of the events affected by *SRSF6* silencing. Thus, in the present study only CE events, identified based on junction-spanning reads, were considered for further analyses. An alternative splice event was considered significant if passing below a false discovery rate < 0.05. Potential hits were further filtered by the absolute change in their "percent spliced-in" (PSI) value ($|\Delta PSI| > 0.05$) as well as a $\log_2$-transformed sum of junction-spanning reads supporting the event >5 (mean between replicates). This process yielded 1,212 significantly alternatively spliced exons.

Exon types were annotated using the software Exon Ontology (Tranchevent et al, 2017) (http://fasterdb.ens-lyon.fr/ExonOntology/) (Fig 3D). The strengths of 3' and 5' splice sites of regulated and non-regulated exons (Fig S3A and B) were calculated based on the

maximum entropy model using the sequence analysis software MaxEntScan (Yeo & Burge, 2004). Non-regulated exons from the same transcripts of the regulated exons were considered for comparison.

For the overlap of SRSF6-regulated CEs and SRSF6 binding (Fig 3B), all binding sites in the alternatively spliced region were taken into account. This was defined as the window from –100-nt of the 5′ splice site of the upstream constitutive exon until +100-nt of the 3′ splice site of the downstream constitutive exon.

### SRSF6 RNA splicing maps

Significantly alternatively spliced exons were grouped by their decreased or increased inclusion based on their ΔPSI values. The 5′ and 3′ splice sites of the regulated and the flanking up- and downstream exons were used as reference point to span a symmetric window of 200-nt. On each nucleotide position in each window, all exons with at least one cross-link event at this position were counted and divided by the total number of exons. The resulting relative SRSF6 cross-link frequency is shown in Fig 3C.

A common problem when integrating orthogonal data with alternatively spliced exons are the differences in the base inclusion level between exons. For example, an exon with increased inclusion upon *SRSF6* KD must show less inclusion in the control situation and thus will show fewer iCLIP cross-link counts (Fig S3C). To deal with this, we compiled a PSI-matched background set as reference for the exons with decreased inclusion, and a second PSI-matched background set for the exons with increased inclusion. These were chosen to match in their distribution of PSI values in control conditions. As a starting point, we used all exons not included in the two significant sets (total background). For each set, we calculated the distribution of PSI values, split it into 5% quantiles and then transferred the quantile boundaries to randomly pick an adjusted background set from the total background. The adjusted background set was subjected to a repeated subsampling approach, such that a PSI-matched background set of equal size as the regulated set was picked 50 times (Fig S3D and E). For each iteration, the relative SRSF6 cross-link frequency was calculated in the same way as described above. We then used the mean and SD of binding profiles on the 50 PSI-matched background sets to calculate a positional z-score as well as a *P*-value.

### Functional annotation and susceptibility genes

Functional enrichment analysis was performed with clusterProfiler (Yu et al, 2012) for Gene Ontology. The genes identified by RNA-seq in control or after *SRSF6* KD with normalized read counts >1 were considered as expressed and therefore used as background. The standard parameters were applied, and an adjusted *P*-value < 0.05 was considered to be statistically significant.

T1D and T2D susceptibility genes were identified from ImmunoBase (www.immunobase.org, accessed on November, 2018), GWAS catalog (https://www.ebi.ac.uk/gwas/, accessed November, 2018), and (Wen & Yang, 2017; Udler et al, 2018). This yielded a total of 102 T1D and 330 T2D susceptibility genes.

### PPI network

The Search Tool for the Retrieval of Interacting Genes version 11.0 (STRING; https://string-db.org/) was used to identify PPI networks (Szklarczyk et al, 2019). The merged list of T1D and T2D susceptibility genes with significant AS events and/or a significant differential expression (with |LFC| > 1) were used as input (n = 29). Only interactions supported by direct experiments were taken into account. A median confidence score >0.4 was considered relevant. The PPI network was further modified in Adobe Illustrator for color coding (Fig S7C).

### Retrieval of Genotype-Tissue Expression (*GTEx*) data sets

The gene read counts of the RNA-Seq GTEx version 8 data set (GTEx_Analysis_2017-06-05_v8_RNASeQCv1.1.9_gene_tpm.gct.gz) and the respective information about the exon-exon junction read counts (GTEx_Analysis_2017-06-05_v8_STARv2.5.3a_junctions.gct.gz) were downloaded from the GTEx Portal (https://gtexportal.org/home/datasets, on 09 May, 2020).

### Transfection of antisense oligonucleotides (ASOs)

The sequence of the antisense oligonucleotide (ASO) (Eurogentec) molecules used in this study are shown in Table S3. All ASOs contained 2′-O-methyl RNA nucleosides and a full-length phosphorothioate backbone. The ASOs were diluted to a final concentration of 200 µM in sterile Tris–EDTA (TE) buffer solution (Promega) and stored at –80°C. EndoC-βH1 cells were transfected for 6 h with 50 µM of each AOS using Lipofectamine 2000 (Thermo Fisher Scientific) following the manufacturer's instructions. An ASO with a scrambled sequence that has no homology in the human genome was used as a negative control (Ziegler et al, 1997). The cells were harvested 48 h after transfection. The concentration of the ASOs was assessed with a NanoDrop spectrophotometer (NanoDrop ND-1000; Thermo Fisher Scientific).

### mRNA extraction, quantitative PCR, and validation of splicing events by RT-PCR

Poly(A)+ mRNA was isolated using the Dynabeads mRNA DIRECT Kit (Invitrogen), according to the manufacturer's instructions. The mRNA molecules were recovered in Tris–HCl elution solution and reverse-transcribed using the Reverse Transcriptase Core kit (Eurogentec), according to the manufacturer's instructions. The quantitative PCR amplification was conducted using IQ SYBR Green Supermix (Bio-Rad) using Rotor-Gene Q (QIAGEN). The PCR product concentration was calculated as copies per µl using the standard curve method (Overbergh et al, 1999), and the gene expression was corrected for the reference gene encoding β-actin. The primers used are listed in Table S7. The observed AS events were validated by RT-PCR using specifically designed primers annealing to the flanking constitutive exons (Table S7). The RT-PCR was conducted using the RedTaq DNA polymerase (Bioline) following the manufacturer's instructions. PCR products were analyzed using the LabChip electrophoretic Agilent 2100 Bioanalyzer system and the DNA 1000 LabChip kit (Agilent Technologies). The molarity of each PCR band corresponding to a specific splice variant was quantified using the 2100 Expert Software (Agilent Technologies), and used to calculate the percentage of inclusion of the target CE (% inclusion = molarity bigger amplicon/(molarity bigger amplicon + molarity smaller amplicon) × 100).

### Statistical analysis

Data are shown as means ± SEM. Significant differences between experimental conditions were determined by paired $t$ test or by ANOVA followed by Bonferroni correction as indicated. $P$-values < 0.05 were considered statistically significant. Statistical tools used for the analysis of iCLIP and RNA-seq are described above.

# Data Availability

RNA sequencing (RNA-seq) data of EndoC-$\beta$H1 cells were previously published (Juan-Mateu et al, 2018) and are available at GEO under accession number GSE98485. SRSF6 iCLIP data have been deposited at GEO under accession number GSE150172. The computational code for the binding site definition, the motif analysis, the AS analysis, and the RNA splicing maps are available with this manuscript (Supplemental Data 1).

# Supplementary Information

# Acknowledgements

The authors are grateful to Isabelle Millard, Anyishaï Musuaya, Nathalie Pachera, Cai Ying, and Manon Depessemier (ULB Center for Diabetes Research) for providing excellent technical support. The authors would like to thank Dr Michaela Müller-McNicoll and all members of the Zarnack and Eizirk groups for fruitful discussions. Funding information: DL Eizirik is funded by Welbio/FRFS (no WELBIO-CR-2019C-04), Belgium; by the Brussels Region (INNOVIRIS BRIDGE grant DiaType), the Innovative Medicines Initiative 2 Joint Undertaking under grant agreement numbers 115797 (INNODIA) and 945268 (INNODIA HARVEST); these Joint Undertakings receive support from the Union's Horizon 2020 research and innovation program and "EFPIA" (European Federation of Pharmaceutical Industries Associations), "JDRF" (Juvenile Diabetes Research Foundation), "The Leona M and Harry B Helmsley Charitable Trust"), and the Dutch Diabetes Research Foundation (project Innovate2CureType1, DDRF; no. 2018.10.002). MI Alvelos was supported by Fonds pour la Formation a la Recherche dans l'Industrie et dans l'Agriculture (FRIA) fellowship from the Fonds de la Recherche Scientifique (FNRS) (reference no. 26410496), and COST: European Cooperation in Science & Technology (COST Action BM1207—Networking towards clinical application of antisense-mediated exon skipping; COST Action CA17103—Delivery of Antisense RNA Therapeutics). K Zarnack is funded by the German Research Foundation (DFG, SFB 902 – B13).

### Author Contributions

MI Alvelos: conceptualization, data curation, formal analysis, validation, investigation, visualization, methodology, and writing—original draft, review, and editing.

M Brüggemann: data curation, software, formal analysis, validation, investigation, visualization, methodology, and writing—original draft, review, and editing.

FXR Sutandy: formal analysis, methodology, and writing—review and editing.

J Juan-Mateu: conceptualization, investigation, methodology, and writing—review and editing.

ML Colli: data curation, investigation, methodology, and writing—review and editing.

A Busch: data curation, software, formal analysis, and writing—review and editing.

M Lopes: formal analysis and writing—review and editing.

Â Castela: validation, methodology, and writing—review and editing.

A Aartsma-Rus: methodology and writing—review and editing.

J König: resources, formal analysis, funding acquisition, methodology, and writing—review and editing.

K Zarnack: conceptualization, resources, data curation, software, formal analysis, supervision, validation, investigation, visualization, methodology, and writing—original draft, review, and editing.

DL Eizirik: conceptualization, resources, data curation, software, formal analysis, supervision, funding acquisition, validation, investigation, visualization, methodology, project administration, and writing—original draft, review, and editing.

### Conflict of Interest Statement

A Aartsma-Rus discloses being employed by Leiden University Medical Center (LUMC) which has patents on exon skipping technology, some of which has been licensed to BioMarin and subsequently sublicensed to Sarepta. As co-inventor of some of these patents A Aartsma-Rus is entitled to a share of royalties. A Aartsma-Rus further discloses being ad hoc consultant for PTC Therapeutics, Sarepta Therapeutics, CRISPR Therapeutics, Summit PLC, Alpha Anomeric, BioMarin Pharmaceuticals Inc., Eisai, Astra Zeneca, Santhera, Audentes, Global Guidepoint and GLG consultancy, Grunenthal, Wave, and BioClinica, having been a member of the Duchenne Network Steering Committee (BioMarin) and being a member of the scientific advisory boards of ProQR, hybridize therapeutics, silence therapeutics, Sarepta therapeutics, and Philae Pharmaceuticals. Remuneration for these activities is paid to LUMC. LUMC also received speaker honoraria from PTC Therapeutics and BioMarin Pharmaceuticals and funding for contract research from Italpharmaco and Alpha Anomeric. Project funding is received from Sarepta Therapeutics. The other authors declare that they have no competing interests.

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
