## [Reviewer comments · Life Science Alliance]

Life Science Alliance

The RNA binding profile of the splicing factor SRSF6 in immortalized human pancreatic β -cells

Maria Alvelos, Mirko Brüggemann, FX Sutandy, Jonàs Juan-Mateu, Maikel Colli, Anke Busch, Miguel Lopes, Ângela Castela, Annemieke Aartsma-Rus, Julian König, Kathi Zarnack, and Decio Eizirik
DOI: <https://doi.org/10.26508/lsa.202000825>

Corresponding author(s): Maria Alvelos, ULB Center for Diabetes Research

Review Timeline:

Submission Date:	2020-06-22
Editorial Decision:	2020-08-06
Revision Received:	2020-10-13
Editorial Decision:	2020-12-07
Revision Received:	2020-12-15
Accepted:	2020-12-15

Scientific Editor: Shachi Bhatt

Transaction Report:

August 6, 2020

Re: Life Science Alliance manuscript #LSA-2020-00825-T

Ms. Maria Ines Alvelos
ULB Center for Diabetes Research
U.L.B. CP618 Route de Lennik 808
Brussels (Anderlecht) 1070
Belgium

Dear Dr. Alvelos,

Thank you for submitting your manuscript entitled "The RNA binding profile of the splicing factor SRSF6 in human pancreatic β -cells" to Life Science Alliance. The manuscript was assessed by expert reviewers, whose comments are appended to this letter.

As you will see, the referees appreciate that you further assess the function of SRSF6 in human pancreatic cells and propose a link to diabetes, but they also raise a number of concerns that would need to be addressed in a revised version.

In particular, all referees find that the conclusions regarding the proposed link of SRSF6 function to diabetes are not sufficiently supported by the experimental data. This concern should be addressed by careful revision of the text, as well as by providing additional data. Both referees #2 (point 1) and #3 (point 6) suggest comparing the effects of the GLIS3 knockdown on alternative splicing of diabetes susceptibility genes to that of the SRSF6 knockdown, and this should be done using the available datasets. In addition, the proposed role for SRSF6 in alternative splicing of LMO7 as a representative candidate should be further strengthened (ref #2 - point 6, 7; ref #3- minor point 2). The referees' questions should be addressed, in particular by providing additional evidence for a role of LMO7 alternative splicing in beta cell function and a direct effect of SRSF6, which may in part be possible by re-analyzing available data.

Please also carefully consider all other points the referees raise and add to the discussion and revise the text accordingly, in particular to ensure that the conclusions are fully supported by the data. If you are able to fully resolve these issues, we will be happy to consider the manuscript further for publication in Life Science Alliance. Therefore we would now like to invite you to prepare and submit a revised version of the manuscript.

In our view these revisions should typically be achievable in around 3 months. However, we are aware that many laboratories cannot function fully during the current COVID-19/SARS-CoV-2 pandemic and therefore encourage you to take the time necessary to revise the manuscript to the extent requested above. We will extend our 'scoping protection policy' to the full revision period required. If you do see another paper with related content published elsewhere, nonetheless contact me immediately so that we can discuss the best way to proceed.

You will be guided to complete the submission of your revised manuscript and to fill in all necessary

information. Please get in touch in case you do not know or remember your login name.

Thank you for this interesting contribution to Life Science Alliance. We are looking forward to receiving your revised manuscript.

Sincerely,

Reilly Lorenz
Editorial Office Life Science Alliance
Meyerhofstr. 1
69117 Heidelberg, Germany
t +49 6221 8891 414
e contact@life-science-alliance.org
www.life-science-alliance.org

B. MANUSCRIPT ORGANIZATION AND FORMATTING:

Reviewer #1 (Comments to the Authors (Required)):

The authors of this study have previously shown that inhibition of the splicing factor SRSF6 affects the function and viability of pancreatic B cells. In this study they performed iCLIP experiments in a human pancreatic B-cell line, and report more than 180,000 binding sites for SRSF6 in more than 8000 genes, many of which overlap with published SRSF6 differentially spliced genes. Given that this number of genes is likely to constitute a major fraction of active genes, it is not unexpected that many cassette binding sites overlap with diabetes GWAS genes. The authors provide experimental validations showing that knockdown of SRSF6 affects splicing of 7/9 Diabetes GWAS genes. They also show they can use antisense oligos to cause exon skipping in one Diabetes GWAS gene, which can theoretically be used counteract the effects of SRSF6.

The manuscript claims that SRSF6 has a role in Diabetes susceptibility, although the data does not really show that SRSF6 is altered in Diabetes mechanisms or susceptibility. However, the iCLIP and knockdown experiments are well executed. The SRSF6 iCLIP findings seem solid and are generally in line with previous iCLIP data, and will provide a useful resource. The results do suggest a potential role of SRSF6 in Diabetes mechanisms.

Major and minor points:

The manuscript is well written, but the claims about diabetes should be toned down (e.g. abstract last sentence, results section page 9)

Figure 3C. The signal in unchanged exons cannot be seen due to the choice of colors.

Figure 4. The plot is useful, but does not clearly illustrate the relationship between differential inclusion and binding. Is there enrichment?

Reviewer #2 (Comments to the Authors (Required)):

The manuscript by Alvelos et al. describes the characterization of the splicing factor SRSF6 in a human pancreatic beta cell line. This group previously reported SRSF6/SRp55 was regulated by GLIS3 and played a role in beta cell survival and function. The current study goes on to characterize SRSF6 RNA binding characteristics in a human beta cell line using several innovative technologies. These experiments identified a candidate SRSF6 consensus binding motif and provided information related to how SRSF6 positioning within a gene influenced splicing events. The study also identified potential SRSF6 targets and partially characterized one of these targets: LMO7. The data is of high quality and the manuscript is clearly written. Overall, the study provides novel information regarding the molecular activity of SRSF6 - and this is the strongest feature of the paper. However, the claims related to the role of SRSF6 in beta cell function and diabetes are overstated. As discussed below, some additional information would improve the quality of the study and strengthen the link to beta cell biology.

Major points:

1. SRSF6 was initially identified as a beta cell target of GLIS3. To strengthen the "beta cell functional" aspect of this study, it would be informative to compare the splicing defects that occur in the GLIS3 KO/KD and SRSF6 KO/KD. This could be done computationally with the existing data sets. Most importantly, were the "diabetes susceptible targets" of SRSF6, including LMO7 mis-spliced in the GLIS3 KO beta cells.
2. On page 5 and in Supplementary figure 1A, the authors show that SRSF6 is expressed at relatively equal levels in EndoC-betaH1 cells, HeLa cells and HEK293 cells. Several published human single cell datasets also show relatively equal levels of SRSF6 across most tissues. Yet on page 15 and in their previous publication (Juan-Mateu, 2018), the authors indicate that SRSF6 is higher in pancreatic islets and EndoC-betaH1 cells. This discrepancy should be reconciled; especially since SRSF6 does not appear to be especially beta cell specific.
3. The optimization of the iCLIP experiments in Figure 1 appear to be primarily optimization of the UV crosslinking step. Since the iCLIP experiments were subsequently performed on 8×10^6 cells under the 300 mJ/cm^2 conditions, the control RNase experiment should be included using these conditions, especially since it will be important to show that the higher level of crosslinking doesn't affect specificity of the interactions.
4. On page 7, the authors state "The majority of AS events harbored one or more SRSF6 binding sites in the alternatively spliced region, indicating that they are direct targets of SRSF6 regulation". A correlation between iCLIP and RNA-Seq data is not sufficient to prove direct targets; this would require showing that mutation of the putative SRSF6 binding site disrupted the splicing event.
5. On page 8, the authors mention SRSF4 is upregulated upon down regulation of SRSF6. Did they test for compensation? Does SRSF4 recognize the same targets or does its activation regulate a new set of targets? - which would complicate the interpretation of altered splicing events directly related to SRSF6 activity. On a related note, the authors also indicate that "SRSF6 induces expression of many other splicing regulators" - this also brings up the potential of direct vs. indirect splicing changes associated with SRSF6 KD and possible compensatory mechanisms.
6. One of the weakest aspects of the study is the link between SRSF6 targets and diabetes/beta cell function. Although many of the genes that were identified have shown up in GWAS studies, the genes listed are not beta cell specific, are widely expressed in many tissues, and have roles in general cellular functions, such as chromosome segregation and cell cycle regulation. LMO7 is also widely expressed in all human tissues. To strengthen the relationship to beta cell function, the authors should expand their analysis of LMO7 alternative splicing. Does the T1D SNP affect splicing of exon 10. Does dysregulation of GLIS3 affect the splicing of exon 10? Does mis-splicing of exon 10 affect beta cell function? The authors present several pieces of correlative data related to LMO7 expression, but do not demonstrate that mis-splicing of LMO7 can affect beta cell function.
7. It is unclear why authors tested the general blockage of LMO7 splicing rather than determining

whether blocking SRSF6 binding sites would alter splicing, which would begin to elucidate the mechanism through which SRSF6 regulates the splicing of exon 10 (and is the main focus of this study). The ASOs were designed to block the splice recognition sites independently of SRSF6.

Minor points:

1. Antisense Oligonucleotides are more commonly abbreviated as ASOs rather than AONs
2. In Figure 1D, there is not enough information describing the binding and flanking regions; what are the yellow boxes?
3. Why did the authors refer to the protein as SRp55 in the previous publication and SRSF6 in this study? They do indicate the alternate names in each publication, but is it strange that they weren't consistent between manuscripts.
4. In the title (and throughout the manuscript) the authors should indicate "EndoC-betaH1 cells" or "immortalized human beta cells" rather than "human beta cells"

Reviewer #3 (Comments to the Authors (Required)):

The manuscript by Alvelos et al is an extension of their previous work regarding GLIS3, a diabetes susceptibility gene encoding the transcription factor that regulates SRSF6, which influences alternative splicing in beta cells. The main findings of this paper nicely complement and extend the author previous work and other work in the field. The main advancements in this paper are the definition of SRSF6-bound transcripts in human beta cells, and the binding site sequence and position in each transcript. These data, in combination a reanalysis of previous transcriptome data from SRSF6 KD EndoC- β H1 cells demonstrate that the majority of transcripts targeted by SRSF6 are alternatively spliced, and that a number of the alternatively spliced transcripts are diabetes susceptibility genes, some of which harbor a SRSF6 binding site. Finally, the authors also find that the SRSF6-splicing result depends on its binding position in exons and the effect of the KD could be attenuated by antisense oligonucleotides spanning the splice sites. The manuscript demonstrates that alternative splicing in beta cells in an additional layer of genetic predisposition to diabetes: one diabetes susceptibility gene through alternative splicing influences a network of diabetes susceptibility genes, which is further expanded through their protein-protein interactions. However, there are some inconsistencies in the binding sequence that requires further clarifications.

Some major and minor concerns and suggestions listed below:

1. In the material and methods, the authors state that the highest scoring de novo motif was a U-rich motif. This is consistent with figure 2C, where AU-rich pentamers were the most enriched at binding site centers. But in the binding sites shown in the CCDC50 example (figure 1D), there are no uridine nucleotides in either of the binding sites. Furthermore, the binding motif was defined as a purine rich motif, and GAA triplets were used to analyze binding site strength, even though U-rich sequences were determined to be in SRSF6 binding sites. The determined binding motif in figure 2F is not an enriched pentamer within the defined 9 nt binding site in figure 2A, neither in exons or in introns. In the intronic binding site, U-rich pentamers were enriched, but this was not taken into account for motif definition. The authors should clarify these discrepancies and why the second hit for motif definition was defined as the SRSF6 binding motif.

2. Suppl. figure 2A shows different data: compared with the middle panel of figure 2A, in the

supplementary figure, AG-rich pentamers are enriched within binding sites, but in figure 2A the same pentamers are enriched only in the regions flanking the binding site. What is the difference between the two plots?

3. From the data, it appears that SRSF6 binds to different sequences in intron and exons, which was demonstrated for both the actual SRSF6 binding site and 20 nt up- and downstream of it. Can the authors elaborate how the same protein would have different binding sites depending on the region of RNA?

4. Regarding suppl. Figure 1D: The iCLIP had no control for unspecific binding, such as non-crosslinked cells or SRSF3 KD/KO cells. Can the authors clarify how they accounted for unspecific binding of SRSF3 in their binding site analysis?

5. Figure 2D and 2E show that two or more consecutive GAA triplets show an increase in binding strength compared to the complement control. It would be possible to strengthen this hypothesis using an RNA pull-down with sequences containing increasing triplicates or SRSF6 targets that have the enriched pentamers flanking the binding site. However, as the strength of binding increases, the number of binding sites decreased. Does this mean that if more GAA triplets are present, SRSF6 has a higher affinity for these sequences so less SRSF6 proteins bind to achieve the same effect?

6. Since the KD of SRSF6 had the effect of alternative splicing, it would be a confirmation if its upstream regulator (GLIS3) KD also affected the splicing of diabetes susceptibility genes as shown in figure 4E.

Minor points.

1. The binding site of SRSF6 is a 9mer, but pentamers are analyzed to determine the binding motif. An explanation of why pentamers were chosen is missing.

2. Is the T1D-associated SNP within the alternatively splices exon10 of LMO7?

3. Can the authors clarify if CDK2 has an SRSF6 binding site? According to figure 4A, it does not, since it is not labeled in dark blue. But the inclusion of the cassette exon is decreased upon SRSF6 KD. Is this due to indirect effects? On figure 4B, STARD10 is not shown to have SRSF6 regulated alternative splicing. It would be important to show the other two exons of the nine that were not validated in this figure, even if there was no difference.

4. Why was CCDC50 chosen as an example in figure 1D? Is there a relevance for beta cells and/or diabetes? While SRSF6 binding was enriched in the CDS compared to introns relative to their respective sizes, the majority of the binding sites were in introns. An example where SRSF6 binds to introns can be included in this figure, especially since the pentamer frequency flanking intronic binding sites is different to that in exons, according to figure 2A

5. Since it is presented in the results, SRSF4 should be labeled on the volcano and scatterplots in suppl. Figure 5A and 5B. Also, the suppl. Figure 4C and 4D contribute very little to the study. Why is this gene relevant for the study, apart from it being another splicing factor? Is SRSF4 also regulated by GLIS3? Why is this gene relevant for beta cells and/or diabetes? Since it does not seem to

contribute much, these data could be removed without taking anything away from main points of the manuscript.

6. In the abstract, the way it is written sounds like the iCLIP was done on SRSF6 KD cells. While this is clarified in the text, the abstract as such can be misleading.

7. Take care of references - on pg. 3 Eizirik, Sammeth et al 2012 are cited, then a couple sentences later, Eizirik et al 2012 is cited for the same paper.

8. In suppl. Figure 1A, why are there no error bars for HEK293 cells?

9. The figure panels should be presented in order in the results (for instance, Figure 1C and 1D should go after 1B, not before).

10. In suppl. Figure 4A, a gene legend above the plot, like in figure 3F, would make it easier to understand where exons and introns are positioned

11. Some of the legends could be better defined. For instance, in the pie chart in figure 1C - what is the number in brackets?

Response to Reviewer Comments

Reviewer #1:

The authors of this study have previously shown that inhibition of the splicing factor SRSF6 affects the function and viability of pancreatic B cells. In this study they performed iCLIP experiments in a human pancreatic B-cell line, and report more than 180,000 binding sites for SRSF6 in more than 8000 genes, many of which overlap with published SRSF6 differentially spliced genes. Given that this number of genes is likely to constitute a major fraction of active genes, it is not unexpected that many cassette binding sites overlap with diabetes GWAS genes. The authors provide experimental validations showing that knockdown of SRSF6 affects splicing of 7/9 Diabetes GWAS genes. They also show they can use antisense oligos to cause exon skipping in one Diabetes GWAS gene, which can theoretically be used counteract the effects of SRSF6.

The manuscript claims that SRSF6 has a role in Diabetes susceptibility, although the data does not really show that SRSF6 is altered in Diabetes mechanisms or susceptibility. However, the iCLIP and knockdown experiments are well executed. The SRSF6 iCLIP findings seem solid and are generally in line with previous iCLIP data, and will provide a useful resource. The results do suggest a potential role of SRSF6 in Diabetes mechanisms.

We are grateful to the Reviewer for the positive comments and relevant suggestions for improvement.

Major and minor points:

The manuscript is well written, but the claims about diabetes should be toned down (e.g. abstract last sentence, results section page 9)

Following the Reviewer's suggestion, we have toned down the claims about the impact of SRSF6 in diabetes throughout the manuscript and made it clearer that our findings were obtained in the cell line EndoC- β H1.

Figure 3C. The signal in unchanged exons cannot be seen due to the choice of colors.

We have changed to stronger colors for the background exons to make the signal more visible.

Figure 4. The plot is useful, but does not clearly illustrate the relationship between differential inclusion and binding. Is there enrichment?

The relation between the differential inclusion and binding is assessed in the RNA maps in **Figure 3C**, in which we show the enrichment of SRSF6 binding on exons that are differentially regulated.

The scatterplots in Figure 4A and 4B show the inclusion level difference of SRSF6-regulated alternative splicing events in T1D and T2D susceptibility genes plotted against binding site strength (\log_2 -transformed PureCLIP score). They additionally provide information about the presence/absence of the identified motif in this binding site (color shading). The plots should thus be seen as a summary of information on these splicing events.

We did not observe an enrichment of stronger binding sites with a particular direction of regulation. A direct comparison of binding site strength between these categories would be

skewed by the differences in relative abundance of the underlying transcript regions, since up- and down-regulated exons show different levels of baseline inclusion under control conditions (**Supplementary Figure 3E**). Since it would be difficult to correct the PureCLIP scores for this bias, we decided against performing such an analysis.

Reviewer #2:

The manuscript by Alvelos et al. describes the characterization of the splicing factor SRSF6 in a human pancreatic beta cell line. This group previously reported SRSF6/SRp55 was regulated by GLIS3 and played a role in beta cell survival and function. The current study goes on to characterize SRSF6 RNA binding characteristics in a human beta cell line using several innovative technologies. These experiments identified a candidate SRSF6 consensus binding motif and provided information related to how SRSF6 positioning within a gene influenced splicing events. The study also identified potential SRSF6 targets and partially characterized one of these targets: LMO7. The data is of high quality and the manuscript is clearly written. Overall, the study provides novel information regarding the molecular activity of SRSF6 - and this is the strongest feature of the paper. However, the claims related to the role of SRSF6 in beta cell function and diabetes are overstated. As discussed below, some additional information would improve the quality of the study and strengthen the link to beta cell biology.

We thank the Reviewer for the positive comments and for considering the present data of high quality. As described below, we have performed new experiments and analyses to address the Reviewer's comments.

Major points:

1. SRSF6 was initially identified as a beta cell target of GLIS3. To strengthen the "beta cell functional" aspect of this study, it would be informative to compare the splicing defects that occur in the GLIS3 KO/KD and SRSF6 KO/KD. This could be done computationally with the existing data sets. Most importantly, were the "diabetes susceptible targets" of SRSF6, including LMO7 mis-spliced in the GLIS3 KO beta cells.

We thank the Reviewer for pointing this out. To address this pertinent question, we used two different approaches.

First, we used RNA-seq data from healthy individuals in the Genotype-Tissue Expression (GTEx) database (version 8, <https://gtexportal.org/>). We selected 127 pancreatic tissue samples and correlated the gene expression of *GLIS3* and *SRSF6* (in transcripts per million, TPM). Consistent with the previous results in rat Ins1E cells, we observed a significant positive correlation between *SRSF6* and *GLIS3* gene expression (Pearson correlation coefficient = 0.58, *P* value < 1.1e-12). The correlation between *GLIS3* and *SRSF6* gene expression is now shown in the **new Figure 5A**.

Proceeding from these data, we attempted to use read counts on exon-exon junctions to calculate the inclusion of the SRSF6-regulated splicing events in the diabetes susceptibility genes *LMO7*, *CDK2*, *CENPO*, *ITGB3BP*, *BCAR1*, *RBM6*, and *STARD10*. Unfortunately, junction coordinates could not be matched for three out of seven splicing events, and only few reads could be retrieved for the remaining events, resulting in high variability between samples. For instance, there was an average of just six junction-spanning reads detecting

either *LMO7* exon 10 inclusion or skipping, precluding reliable estimates of exon inclusion. We therefore decided not to pursue the GTEx data analysis at the level of individual splicing events.

Second, in order to investigate the splicing events in more detail, we tested the effect of *GLIS3* knockdown in the EndoC- β H1 cell line. We depleted *GLIS3* using specific siRNAs and confirmed its reduced expression by qPCR and staining for apoptotic cells, supporting that *GLIS3* depletion contributes to EndoC- β H1 apoptosis. Consistent with previous results in rat cells and our GTEx analysis (see above), *GLIS3* depletion led to *SRSF6* downregulation. Importantly, *GLIS3* depletion also induced splicing changes in two of the *SRSF6*-regulated T1D and T2D susceptibility genes, namely *LMO7* and *ITGB3BP*. Three of the remaining genes showed a trend in the expected direction (*CENPO*, *CDK2*, *RBM6*) while one of them, *STARD10*, went in the opposite direction. One reason may be that reduction of *SRSF6* mRNA levels upon *GLIS3* depletion, which goes down by ~20%, is milder compared to the direct *SRSF6* KD. Another reason is that *GLIS3* has a broad impact on b-cell phenotype and survival, which most probably goes beyond its effects on *SRSF6* regulation. The new data have been included in the **new Figure 5B-E** and the **new Supplementary Figure 6C, D** and is commented upon on in Results, page 9, paragraph 1 (line 266).

2. On page 5 and in Supplementary figure 1A, the authors show that *SRSF6* is expressed at relatively equal levels in EndoC-betaH1 cells, HeLa cells and HEK293 cells. Several published human single cell datasets also show relatively equal levels of *SRSF6* across most tissues. Yet on page 15 and in their previous publication (Juan-Mateu, 2018), the authors indicate that *SRSF6* is higher in pancreatic islets and EndoC-betaH1 cells. This discrepancy should be reconciled; especially since *SRSF6* does not appear to be especially beta cell specific.

We agree with the Reviewer's comment. The measurements in our previous publication did not provide sufficient evidence to draw reliable conclusions on expression differences of *SRSF6* between cell lines and tissues due to different sample preparation protocols. We therefore decided to remove this statement from the current manuscript.

In order to substantiate our new cell line measurements, we increased the number of replicate experiments from EndoC- β H1, HeLa, and HEK293 cells and measured the *SRSF6* mRNA levels by qRT-PCR. The *SRSF6* mRNA is similarly expressed in the three cell types as stated on page 5 of the manuscript, supporting the role of *SRSF6* as a ubiquitously expressed splicing regulator.

The new qPCR data are included in the new **Supplementary Figure 1A**.

3. The optimization of the iCLIP experiments in Figure 1 appear to be primarily optimization of the UV crosslinking step. Since the iCLIP experiments were subsequently performed on 8×10^6 cells under the 300 mJ/cm² conditions, the control RNase experiment should be included using these conditions, especially since it will be important to show that the higher level of crosslinking doesn't affect specificity of the interactions.

We tested a panel of different conditions and combinations during the optimization of the iCLIP experiment. We agree that the high-RNase condition was applied to the correct number of cells (8×10^6), but not in combination with the finally used UV dose of 300 mJ/cm². We consider it unlikely that the specificity drastically changed in response to the

UV dose. Unfortunately, due to the very large number of cells required and limited activities in the lab by the COVID 19 pandemic, we are not able to repeat the iCLIP experiment.

4. On page 7, the authors state "The majority of AS events harbored one or more SRSF6 binding sites in the alternatively spliced region, indicating that they are direct targets of SRSF6 regulation". A correlation between iCLIP and RNA-Seq data is not sufficient to prove direct targets; this would require showing that mutation of the putative SRSF6 binding site disrupted the splicing event.

The Reviewer's comment is well taken. We agree that the presence of SRSF6 binding sites does not necessarily mean that this represents functional binding. We now acknowledge this limitation in the respective sentence on page 7.

5. On page 8, the authors mention SRSF4 is upregulated upon down regulation of SRSF6. Did they test for compensation? Does SRSF4 recognize the same targets or does its activation regulate a new set of targets? - which would complicate the interpretation of altered splicing events directly related to SRSF6 activity. On a related note, the authors also indicate that "SRSF6 induces expression of many other splicing regulators" - this also brings up the potential of direct vs. indirect splicing changes associated with SRSF6 KD and possible compensatory mechanisms.

As pointed out by the Reviewer, we find that *SRSF6* KD affects the expression of *SRSF4*. This compensatory mechanism is well-known for SR proteins (Saltzman AL, Genes & Development 2011, Pandit S, Molecular Cell 2013). The work by Müller-McNicoll M, Genes & Development 2016 shows that exon binding of SRSF4 and SRSF6 is highly correlated (Müller-McNicoll, Supplementary Figure 4E, left panel, shown below) and that both proteins display very similar *in vivo* binding motifs (Müller-McNicoll, Supplementary Figure 4E, right panel, shown below), suggesting that SRSF6 and SRSF4 may regulate similar targets. However, despite their overall similarity, the individual SR proteins can have distinct functions (Müller-McNicoll M, Genes & Development 2016).

Müller-McNicoll M, Genes & Development 2016, Supplementary Figure 4E: Left panel: Hierarchical clustering, using distance correlation and Spearman rank correlation of exon cobinding showing SR protein pairs often binding to the same exons. Right panel: *In vivo* binding motifs of individual SR proteins derived through analysis of enriched octamers surrounding significant cross-link sites in all transcript regions.

It is indeed probable that we may not identify the full spectrum of SRSF6-regulated splicing events due to potential partial compensatory effects by SRSF4. One way to directly address this would be to perform a double-KD of both proteins, but this would likely be detrimental to the cells since individual KDs are already difficult to obtain. Moreover, it is important to highlight that in β -cells, *SRSF6* KD alone is sufficient to trigger cellular dysfunction and apoptosis, indicating that SRSF4 cannot fully compensate here.

Along similar lines, we show that *SRSF6* depletion changes the expression of multiple splicing regulators (Supplementary Figure 5C). As in the case of SRSF4, they may influence the observed alternative splicing changes. We agree that it is important to show this

and make readers aware of these potential confounding effects. In fact, the cross-regulation among splicing factors is not unique to our system, but has been observed in many studies before (e.g., Turunen JJ, RNA 2013; Brooks AN, Genome Research 2015; Lareau LF, Molecular Biology and Evolution 2015). We now acknowledge the possibility or confounding effects from cross-regulation of SR proteins and other splicing regulators in the text, page 8, line 229.

6. One of the weakest aspects of the study is the link between SRSF6 targets and diabetes/beta cell function. Although many of the genes that were identified have shown up in GWAS studies, the genes listed are not beta cell specific, are widely expressed in many tissues, and have roles in general cellular functions, such as chromosome segregation and cell cycle regulation. LMO7 is also widely expressed in all human tissues. To strengthen the relationship to beta cell function, the authors should expand their analysis of LMO7 alternative splicing. Does the T1D SNP affect splicing of exon 10. Does dysregulation of GLIS3 affect the splicing of exon 10? Does mis-splicing of exon 10 affect beta cell function? The authors present several pieces of correlative data related to LMO7 expression, but do not demonstrate that mis-splicing of LMO7 can affect beta cell function.

The function of LMO7 in pancreatic β -cells remains to be clarified. Multiple roles have been described in other cell types, including adherens junction assembly (Ooshio T, Journal of Biological Chemistry 2004), cell migration (Hu Q, Molecular Cell Biology 2011) and gene transcription (Holaska JM, Human Molecular Genetics 2006). A downstream analysis on LMO7's function in β -cells and the impact of LMO7 mis-splicing offers an interesting entry point for future research, but is beyond the scope of this study.

In order to address the Reviewer's questions, we performed the following analyses:

First, we tried to use the *in silico* prediction tools Alamut Visual (version 2.15, Interactive Biosoftware, Rouen, France; www.interactive-biosoftware.com) and SpliceAI (Jaganathan K, Cell 2019) to evaluate the putative impact of the risk SNP in the LMO7 gene on alternative splicing of LMO7 exon 10. However, no splicing modulation could be predicted, possibly due to the deep intronic location of the risk SNP.

Second, we directly tested the impact of GLIS3 depletion on the splicing of SRSF6 target genes. These experiments confirmed that SRSF6 mRNA levels went down after GLIS3 KD. Importantly, the alternative splicing of LMO7 and ITGB3BP responded to the GLIS3 KD in the same direction as predicted from our SRSF6 KD experiments. The remaining splicing events were mildly affected, but did not reach significance, possibly because the GLIS3 KD resulted in a milder depletion of SRSF6 mRNA levels compared to direct SRSF6 KD.

The experiments on the GLIS3 KD are shown in the **new Figure 5** and described on page 9. These include the qPCR and apoptosis measurements for validation of the GLIS3 KD, as well as measurements of SRSF6 mRNA levels and alternative splicing in LMO7 and ITGB3BP. The remaining splicing events are shown in the **new Supplementary Figure 6D, E**.

7. It is unclear why authors tested the general blockage of LMO7 splicing rather than determining whether blocking SRSF6 binding sites would alter splicing, which would begin to elucidate the mechanism through which SRSF6 regulates the splicing of exon 10 (and is

the main focus of this study). The ASOs were designed to block the splice recognition sites independently of SRSF6.

The Reviewer's comment is well taken. Our rationale for the ASO design was that from a therapeutic perspective, an ASO targeting an SRSF6 binding site would mimic the pathological situation of reduced *SRSF6* expression. Instead, we wanted to demonstrate that ASOs can be used to restore the downstream splicing defects in the SRSF6 targets. The experiment should thus be seen as a proof of principle that splicing modulation via ASOs can be achieved in human β -cells.

We agree with the Reviewer that it would be nice to use ASOs to directly manipulate SRSF6 binding at individual binding sites to test their impact on splicing regulation. However, designing such ASOs is not trivial and would most likely require an oligonucleotide walk along the entire region. We therefore decided not to pursue this approach for this manuscript.

Minor points:

1. Antisense Oligonucleotides are more commonly abbreviated as ASOs rather than AONs

We have changed the text and figures accordingly.

2. In Figure 1D, there is not enough information describing the binding and flanking regions; what are the yellow boxes?

We apologize for the lack of detail in the figure legend. We have now added a more detailed description of Figure 1D.

3. Why did the authors refer to the protein as SRp55 in the previous publication and SRSF6 in this study? They do indicate the alternate names in each publication, but is it strange that they weren't consistent between manuscripts.

Due to historical reasons, most SR proteins carry multiple names. Our decision to refer to the protein as SRSF6 in the present manuscript follows the unifying nomenclature for SR proteins proposed by James Manley and Adrian Krainer in 2010 (Manley JL, *Genes & Development* 2010). SRSF6 is also the official gene name according to the HUGO Gene Nomenclature Committee (<https://www.genenames.org/>). We apologize for the inconsistency with the previous publication.

4. In the title (and throughout the manuscript) the authors should indicate "EndoC-betaH1 cells" or "immortalized human beta cells" rather than "human beta cells".

As suggested by the Reviewer, this correction has been added to the manuscript.

Reviewer #3:

The manuscript by Alvelos et al is an extension of their previous work regarding GLIS3, a diabetes susceptibility gene encoding the transcription factor that regulates SRSF6, which influences alternative splicing in beta cells. The main findings of this paper nicely complement and extend the author previous work and other work in the field. The main advancements in this paper are the definition of SRSF6-bound transcripts in human beta

cells, and the binding site sequence and position in each transcript. These data, in combination a reanalysis of previous transcriptome data from SRSF6 KD EndoC- β H1 cells demonstrate that the majority of transcripts targeted by SRSF6 are alternatively spliced, and that a number of the alternatively spliced transcripts are diabetes susceptibility genes, some of which harbor a SRSF6 binding site. Finally, the authors also find that the SRSF6-splicing result depends on its binding position in exons and the effect of the KD could be attenuated by antisense oligonucleotides spanning the splice sites. The manuscript demonstrates that alternative splicing in beta cells in an additional layer of genetic predisposition to diabetes: one diabetes susceptibility gene through alternative splicing influences a network of diabetes susceptibility genes, which is further expanded through their protein-protein interactions. However, there are some inconsistencies in the binding sequence that requires further clarifications.

We are grateful to the Reviewer for the detailed comments and suggestions for improvement, and for pointing that the main findings of the paper nicely complement and extend our previous work and other work in the field.

Some major and minor concerns and suggestions listed below:

1. In the material and methods, the authors state that the highest scoring de novo motif was a U-rich motif. This is consistent with figure 2C, where AU-rich pentamers were the most enriched at binding site centers. But in the binding sites shown in the *CCDC50* example (figure 1D), there are no uridine nucleotides in either of the binding sites. Furthermore, the binding motif was defined as a purine rich motif, and GAA triplets were used to analyze binding site strength, even though U-rich sequences were determined to be in SRSF6 binding sites. The determined binding motif in figure 2F is not an enriched pentamer within the defined 9 nt binding site in figure 2A, neither in exons or in introns. In the intronic binding site, U-rich pentamers were enriched, but this was not taken into account for motif definition. The authors should clarify these discrepancies and why the second hit for motif definition was defined as the SRSF6 binding motif.

The enrichment of uridines is a well-known phenomenon resulting from the uridine bias of UV crosslinking (Sugimoto Y, *Genome Biology*, 2012; Haberman N, *Genome Biology* 2017; Chakrabarti A, *Annual Reviews of Biomedical Data Science* 2018). Thus, the binding sites detected by peak calling on the iCLIP data reflect a combination of the RNA binding preference of the protein and the UV crosslinking bias, such that the highest iCLIP signal is often found at a U-rich position in the immediate neighborhood of the actual binding site. Due to this observation, many motif definition approaches exclude the sequence underlying the (i)CLIP peaks and only focus on the flanking regions.

In our data, the uridine bias of UV crosslinking is illustrated in the metaprofile shown in **Figure 2C** which identifies a needle of UUUUU and AUUUU in the binding site centers (i.e. the site of highest SRSF6 crosslinking), while the GAA-containing pentamers are enriched in a broader peak around the binding sites. It is also reflected in the example of SRSF6 binding in *CCDC50* shown in **Figure 1D**, in which one binding site harbors TTT and the second one carries four T's in the immediate binding site region. Importantly, both binding sites harbor the proposed recognition motif in their close vicinity.

We now explicitly reference three publications for further information on the uridine bias of UV crosslinking and the detection of RNA binding motifs in this context:

Sugimoto, Y., König, J., Hussain, S., Zupan, B., Curk, T., Frye, M. & Ule, J. (2012) Analysis of CLIP and iCLIP methods for nucleotide-resolution studies of protein-RNA interactions. *Genome Biol* 13(8):R67. <https://doi.org/10.1186/gb-2012-13-8-r67>

Haberman, N., Huppertz, I., Attig, J., König, J., Wang, Z., Hauer, C., et al. (2017). Insights into the design and interpretation of iCLIP experiments. *Genome Biology* 18, 7. <http://doi.org/10.1186/s13059-016-1130-x>

Chakrabarti, A. M., Haberman, N., Praznik, A., Luscombe, N. M., & Ule, J. (2018). Data Science Issues in Studying Protein–RNA Interactions with CLIP Technologies. *Annual Review of Biomedical Data Science* 1, 235–261. <http://doi.org/10.1146/annurev-biodatasci-080917-013525>

2. Suppl. figure 2A shows different data: compared with the middle panel of figure 2A, in the supplementary figure, AG-rich pentamers are enriched within binding sites, but in figure 2A the same pentamers are enriched only in the regions flanking the binding site. What is the difference between the two plots?

We apologize for the inconsistent labeling of the scatter plot in Supplementary Figure 2A. Figure 2A shows the pentamer frequencies separately for three regions, i.e. the 9-nt binding sites as well as 20-nt of flanking region to either side. In contrast, Supplementary Figure 2A showed a summarized version, in which pentamer frequencies were counted over the entire 49-nt window.

In order to avoid redundancy and confusion, we have removed Supplementary Figure 2A in the revised manuscript.

3. From the data, it appears that SRSF6 binds to different sequences in intron and exons, which was demonstrated for both the actual SRSF6 binding site and 20 nt up- and downstream of it. Can the authors elaborate how the same protein would have different binding sites depending on the region of RNA?

We indeed observed different pentamers to be enriched in the SRSF6 binding sites in introns and CDSs. We think that most likely this observation does not reflect a different binding specificity of the protein, but rather differences in sequence composition in distinct regions of the transcripts. Coding regions underlie higher sequence constraints than the intervening intronic sequences, in particularly when moving away from the exons, and thus are less likely to harbor extended uridine stretches (UUU encodes for phenylalanine). In addition, due to the shorter half-life of intronic compared to exonic transcript regions, only well-crosslinking binding sites in introns will reach a similar signal intensity as binding sites in exons. We believe that these two aspects may cause the increased frequency of U-rich pentamers in intronic binding sites, which in turn masks the actual GAA-rich motif. Nevertheless, GAA-rich pentamers are still enriched also in introns, supporting that this motif is recognized by SRSF6 in both regions.

We now point to the differences in sequence composition between introns and exons as a possible explanation for this observation on page 6, paragraph 2 (line 162).

4. Regarding suppl. Figure 1D: The iCLIP had no control for unspecific binding, such as non-crosslinked cells or SRSF3 KD/KO cells. Can the authors clarify how they accounted for unspecific binding of SRSF3 in their binding site analysis?

We agree with the Reviewer that unspecific RNA binding of an RNA-binding protein is an important aspect to consider in binding analysis. A no-UV, knockout or no-antibody sample allows to control for unspecific bands appearing during the iCLIP purification. However, these controls are often not proceeded to high-throughput sequencing, since they are expected to require high amplification and to produce limited data for later analysis. Moreover, they will not capture the unspecific RNA binding intrinsic to the RNA-binding protein of interest, as this will also be lost in these controls.

In order to account for background from unspecific binding, we used the peak calling algorithm PureCLIP (Krakau S, Genome Biology 2017). PureCLIP trains a hidden Markov model (HMM) on the data which allows to incorporate non-specific background, UV crosslinking biases and other confounding factors which are inherent to iCLIP/eCLIP datasets. For more details on the procedure and its benchmarking, please refer to the associated publication:

Krakau, S., Richard, H., & Marsico, A. (2017). PureCLIP: capturing target-specific protein-RNA interaction footprints from single-nucleotide CLIP-seq data. *Genome Biology*, 18(1), 240. <http://doi.org/10.1186/s13059-017-1364-25>.

Figure 2D and 2E show that two or more consecutive GAA triplets show an increase in binding strength compared to the complement control. It would be possible to strengthen this hypothesis using an RNA pull-down with sequences containing increasing triplicates or SRSF6 targets that have the enriched pentamers flanking the binding site. However, as the strength of binding increases, the number of binding sites decreased. Does this mean that if more GAA triplets are present, SRSF6 has a higher affinity for these sequences so less SRSF6 proteins bind to achieve the same effect?

Indeed, our analyses indicate that two or more consecutive GAA triplets are associated with increased SRSF6 binding strength. As suggested by the Reviewer, this could be addressed experimentally by RNA pull-down experiments using RNA baits with different numbers of GAA triplets, and also directly in EMSAs or other *in vitro* binding assays. However, such *in vitro* experiments are not trivial, in part due to the stickiness of SR proteins and the difficulties to obtain recombinant proteins. Moreover, it is important to keep in mind that while the GAA triplets clearly associate with stronger SRSF6 binding in our analysis, we also detect widespread RNA binding in absence of the motif. It is unclear whether the quantitative reproducibility of RNA pull-downs followed e.g. by mass spectrometry would be sufficient to resolve the proposed differences in SRSF6 affinity. We therefore decided not to pursue these experiments.

The second questions refers the number of binding sites that are associated with a given number of triplets. As the Reviewer correctly points out, we observed that the binding site strength goes up with increasing GAA triplet numbers, but the absolute number of binding sites in these categories goes down. When looking at the absolute number of binding sites, one needs to keep in mind that an increase in the triplet number increases the length of the total sequence requirement, thus making it less likely to appear by chance. We think that this is a more likely cause for the reduction in binding sites with more GAA triplets, rather than a reduced requirement due to the higher affinity of SRSF6. The control triplet UUC nicely exemplifies this effect by showing a strong decrease at five or more repetitions of the triplet. Compared to UUC, the GAA triplet shows a much slower decay, possibly underpinning the relevance of this pattern.

In order to address this in more detail, we counted the number of repetitions for all 64 possible triplets. Indeed, we find that GAA and AGA show a much slower decay than most other triplets, while increased UUC numbers are equally sparse as most other triplets.

The triplet analysis has now been included as the new **Supplementary Figure 2C**.

6. Since the KD of SRSF6 had the effect of alternative splicing, it would be a confirmation if its upstream regulator (GLIS3) KD also affected the splicing of diabetes susceptibility genes as shown in figure 4E.

We thank the Reviewer for this excellent suggestion. In response to this comment, we performed the analyses and experiments described below.

First, we used RNA-seq data from healthy individuals in the Genotype-Tissue Expression (GTEx) database (version 8, <https://gtexportal.org/>). We selected 127 pancreatic tissue samples and correlated the gene expression of *GLIS3* and *SRSF6* (in transcripts per million, TPM). Consistent with the previous results in rat Ins1E cells, we observed a significant positive correlation between *SRSF6* and *GLIS3* gene expression (Pearson correlation coefficient = 0.58, *P* value < 1.1e-12). The correlation between *GLIS3* and *SRSF6* gene expression is now shown in the **new Figure 5A**.

Proceeding from these data, we attempted to use read counts on exon-exon junctions to calculate the inclusion of the SRSF6-regulated splicing events in the diabetes susceptibility genes *LMO7*, *CDK2*, *CENPO*, *ITGB3BP*, *BCAR1*, *RBM6*, and *STARD10*. Unfortunately, junction coordinates could not be matched for three out of seven splicing events, and only few reads could be retrieved for the remaining events, resulting in high variability between samples. For instance, there was an average of just six junction-spanning reads detecting either *LMO7* exon 10 inclusion or skipping, precluding reliable estimates of exon inclusion. We therefore decided not to pursue the GTEx data analysis at the level of individual splicing events.

Second, in order to investigate the splicing events in more detail, we tested the effect of *GLIS3* knockdown in the EndoC- β H1 cell line. We depleted *GLIS3* using specific siRNAs and confirmed its reduced expression by qPCR and staining for apoptotic cells, supporting that *GLIS3* depletion contributes to EndoC- β H1 apoptosis. Consistent with previous results in rat cells and our GTEx analysis (see above), *GLIS3* depletion led to *SRSF6* downregulation. Importantly, *GLIS3* depletion also induced splicing changes in two of the SRSF6-regulated T1D and T2D susceptibility genes, namely *LMO7* and *ITGB3BP*. Three of the remaining genes showed a trend in the expected direction (*CENPO*, *CDK2*, *RBM6*) while one of them, *STARD10*, went in the opposite direction. One reason may be that reduction of *SRSF6* mRNA levels upon *GLIS3* depletion, which goes down by ~20%, is milder compared to the direct *SRSF6* KD. Another reason is that *GLIS3* has a broad impact on β -cell phenotype and survival, which most probably goes beyond its effects on SRSF6 regulation. The new data have been included in the **new Figure 5B-E** and the **new Supplementary Figure 6C, D** and is commented upon on in Results, page 9, line 266.

Minor points.

1. The binding site of SRSF6 is a 9mer, but pentamers are analyzed to determine the binding motif. An explanation of why pentamers were chosen is missing.

We thank the Reviewer for the comment which we try to clarify below.

It is important to see that the width of the binding site, i.e. the contact region between the RNA-binding protein and the RNA which could be UV crosslinked, does not directly depend on the length of the recognized motif. The recognized sequence is usually rather short and/or degenerate (Dominguez D, Mol Cell, 2018). Also, it does not necessarily lie within the detected iCLIP peak, as explained in more detail above.

For the analysis of sequence composition, the optimal k -mer size depends on the expected size of the recognized motif. Generally, too short k -mers, such as dinucleotides, will show high background frequencies everywhere which are prone to mask the signal, while too long k -mers will dilute the signal. For instance, in the case of a 4-nt recognition motif, a search with hexamers will split the signal into multiple overlapping hits, thereby impairing the observable enrichment. Pentamers are thus a commonly used choice which works well for many RNA-binding proteins (Dominguez D, Mol Cell, 2018). In our case, the motif that is predicted to be recognized by SRSF6 (**Figure 2F**) can be nicely captured in the enriched pentamers (**Figure 2A**).

2. Is the T1D-associated SNP within the alternatively splices exon10 of LMO7?

The risk SNP for T1D diabetes resides in intron 3 in the *LMO7* gene, far away from the next exons. It is important to note that the distal regulatory effects on alternative splicing from deep intronic positions have been described before (Zarnack K, Cell 2013; Shao C, Nature Structural and Molecular Biology 2014). However, it is considerably more difficult to predict such effects compared to immediate splice site mutations etc.

As explained in more detail in the response to Major Comment (6) of Reviewer #2, we tried to use the *in silico* prediction tools Alamut Visual (version 2.15, Interactive Biosoftware, Rouen, France; www.interactive-biosoftware.com) and SpliceAI (Jaganathan K, Cell 2019) to evaluate the putative impact of the risk SNP in the *LMO7* gene on alternative splicing of *LMO7* exon 10. However, no splicing modulation could be predicted, possibly due to the deep intronic location of the risk SNP. We therefore cannot make a reliable conclusion about its impact on *LMO7* alternative splicing. This will be an interesting question for future studies.

3. Can the authors clarify if CDK2 has an SRSF6 binding site? According to figure 4A, it does not, since it is not labeled in dark blue. But the inclusion of the cassette exon is decreased upon SRSF6 KD. Is this due to indirect effects? On figure 4B, STARD10 is not shown to have SRSF6 regulated alternative splicing. It would be important to show the other two exons of the nine that were not validated in this figure, even if there was no difference.

We apologize for the misleading visualization which apparently caused confusion.

In **Figure 4A, B**, we labeled all splicing events that are associated with at least one SRSF6 binding sites in our iCLIP data. The shading of the label in light or dark blue indicates whether this binding site carries the proposed SRSF6 recognition motif (**Figure 2F**). This means that the regulated alternative splicing event in *CDK2* is associated with an SRSF6 binding site without motif.

STARD10 is not labeled in Figure 4B because it is not immediately associated with an SRSF6 binding site in our analysis. The reason lies in the stringent filters that we apply during binding site definition regarding the overlap with multiple annotated genes. In the case of *STARD10*, the open reading frame overlaps with a transcript from a neighboring gene, with both being labeled as high-confidence annotation. The binding sites overlapping with both annotations were therefore excluded from further analysis.

Finally, following the Reviewer's suggestion, we now show the qRT-PCR results for the two alternative splicing events in *GIPR* and *HMGB* that were picked up in our RNA-seq data analysis but could not be validated in qRT-PCR experiments.

The respective plots were included in the **new Supplementary Figure 6A, B**.

4. Why was *CCDC50* chosen as an example in figure 1D? Is there a relevance for beta cells and/or diabetes? While SRSF6 binding was enriched in the CDS compared to introns relative to their respective sizes, the majority of the binding sites were in introns. An example where SRSF6 binds to introns can be included in this figure, especially since the pentamer frequency flanking intronic binding sites is different to that in exons, according to figure 2A

The genome browser view of SRSF6 binding in *CCDC50* is shown in the very first figure to illustrate the binding pattern that we detect in the iCLIP data. The gene has no relevance in β -cells or diabetes. However, it nicely shows the specificity of SRSF6 binding in exons, illustrated by the increased number of uridines within the immediate binding site and the presumed GAA-type recognition motif in close vicinity. The manuscript harbors an additional genome browser view of the SRSF6-regulated alternative splicing target *LMO7* which is a diabetes susceptibility genes and harbors both intronic and exonic SRSF6 binding sites. We therefore did not include further example genes in the revised manuscript.

5. Since it is presented in the results, SRSF4 should be labeled on the volcano and scatterplots in suppl. Figure 5A and 5B. Also, the suppl. Figure 4C and 4D contribute very little to the study. Why is this gene relevant for the study, apart from it being another splicing factor? Is SRSF4 also regulated by GLIS3? Why is this gene relevant for beta cells and/or diabetes? Since it does not seem to contribute much, these data could be removed without taking anything away from main points of the manuscript.

We appreciate the Reviewer's feedback on the visualization.

The main purpose of Supplementary Figure 5A, B is to show the overall expression changes and their relationship with alternative splicing events. Since the differential expression changes of T1D/T2D candidate genes as well as of SR proteins and other splicing regulators are shown in more detail in Supplementary Figure 7A, B and Supplementary Figure 5C, respectively, we now removed all labels from **Supplementary Figure 5B** to avoid redundancy.

The apparent cross-regulation of SRSF6 and SRSF4 is interesting in the light of the overlapping but distinct functionalities of SR proteins (Müller-McNicoll M, Genes & Development 2016). As outlined in more detail in the response to Major Comment (5) of Reviewer #2, the work by Müller-McNicoll M et al. shows that exon binding of SRSF4 and SRSF6 is highly correlated and that both proteins display very similar *in vivo* binding motifs, suggesting that they could partially compensate for each other. We think that this is important to keep in mind, even though we have previously shown that *SRSF6* depletion alone is

sufficient to trigger increased apoptosis in human β -cells, implying that SRSF4 cannot fully compensate here (Juan-Mateu J, Diabetes 2018).

Along similar lines, we show more generally that *SRSF6* depletion changes the expression of multiple splicing regulators (**Supplementary Figure 5C**). As in the case of SRSF4, they may influence the observed alternative splicing changes. We believe that it is important to show and make readers aware of such potential confounding effects. In fact, the cross-regulation among splicing factors is not unique to our system, but has been observed in many studies before (e.g., Turunen JJ, RNA 2013; Brooks AN, Genome Research 2015; Lareau LF, Molecular Biology and Evolution 2015).

6. In the abstract, the way it is written sounds like the iCLIP was done on SRSF6 KD cells. While this is clarified in the text, the abstract as such can be misleading.

We changed the abstract accordingly.

7. Take care of references - on pg. 3 Eizirik, Sammeth et al 2012 are cited, then a couple sentences later, Eizirik et al 2012 is cited for the same paper.

We thank the Reviewer for spotting this inconsistency. We carefully checked all references throughout the manuscript and adapted them to the Science Alliance style.

8. In suppl. Figure 1A, why are there no error bars for HEK293 cells?

We thank the Reviewer for pointing this out. In response to this comment and also to Major Comment (2) of Reviewer #2, we now increased the number of replicates for all cell lines shown in Supplementary Figure 1A.

Supplementary Figure 1A has been exchanged to show the new measurements.

9. The figure panels should be presented in order in the results (for instance, Figure 1C and 1D should go after 1B, not before).

We carefully checked the order of all panels in the final figures of the revised version.

10. In suppl. Figure 4A, a gene legend above the plot, like in figure 3F, would make it easier to understand where exons and introns are positioned

We changed the figure as suggested by the Reviewer.

11. Some of the legends could be better defined. For instance, in the pie chart in figure 1C - what is the number in brackets?

We thank the Reviewer for pointing this out. The pie chart in **Figure 1C** shows the distribution of SRSF6 binding sites per transcript region on protein-coding genes. The numbers show the percentage of binding sites and in brackets the absolute number of binding sites. The information has been added to the legend.

In response to the Reviewer's comment, we carefully revised all figure legends to ensure that all elements are accurately defined.

December 7, 2020

RE: Life Science Alliance Manuscript #LSA-2020-00825-TR

Ms. Maria Ines Alvelos
ULB Center for Diabetes Research
U.L.B. CP618 Route de Lennik 808
Brussels, Brussels (Anderlecht) 1070
Belgium

Dear Dr. Alvelos,

Thank you for submitting your revised manuscript entitled "The RNA binding profile of the splicing factor SRSF6 in immortalized human pancreatic β -cells". We would be happy to publish your paper in Life Science Alliance pending final revisions necessary to meet our formatting guidelines.

We understand the concerns about lack of functional information and absence of direct SRSF6 binding data in the revised manuscript. However, given that all the points raised in the previous LSA decision letter were sufficiently addressed, we consider this manuscript to be ready for publication, barring some formatting related edits.

Along with the points listed below, please also attend to the following,

- please add ORCID ID for corresponding author-you should have received instructions on how to do so
- please use the [10 author names, et al.] format in your references (i.e. limit the author names to the first 10)
- please provide source data (unedited original images) for gels in Figure 5E, Figure 6F, Figure S6

A. FINAL FILES:

-- High-resolution figure, supplementary figure and video files uploaded as individual files: See our detailed guidelines for preparing your production-ready images, <https://www.life-science->

alliance.org/authors

B. MANUSCRIPT ORGANIZATION AND FORMATTING:

Sincerely,

Shachi Bhatt, Ph.D.
Executive Editor
Life Science Alliance
<https://www.lsjournal.org/>
Tweet @SciBhatt @LSAJournal

Reviewer #2 (Comments to the Authors (Required)):

This is a revised submission of a study evaluating the role of Srsf6 in a human beta cell line. The authors have partially addressed the concerns related to the previous version; primarily showing a correlation between splicing changes in the Glis3 KD vs the Srsf6 KD. Many of the other concerns were not addressed experimentally, but the authors did remove some findings that were not supported by the data and toned down many of their over-statements. The study remains strong in its report of Srsf6 iCLIP findings and identification of potential islet beta cell targets; however functional information is still lacking and the authors failed to confirm direct Srsf6 binding, which weakens the study. Furthermore, it is a bit concerning that disputed data and unsupported statements were removed rather than substantiated.

Reviewer #3 (Comments to the Authors (Required)):

In their revised manuscript, Alvelos et al. replied to all of the points of the reviewers, implemented most of the suggestions and added some additional experiments that were requested. The information regarding the binding site definition in the text has improved, while questions regarding the SNP in LMO7 could not be assessed. We understand the limitations in lab work in the middle of the Corona pandemic and appreciate the efforts that the authors took to complete the manuscript. In reference to the additional experiments, Figures 5A and 5C do not add much. The knockdown of GLIS3 in the EndoC cells resulted only in some of the splicing effects that were seen upon knockdown of SRSF6. Among these changes, the authors propose a plausible explanation for three of the genes that were similarly affected in SRSF6 depleted cells, but not for STARD10, since the splicing modifications were in the opposite direction.

The main limitation of this study remains whether SRSF6 mediated mis-splicing of GWAS diabetes risk genes affects beta cell function, which was still not addressed in the revised manuscript. While this can be postulated based on previous authors' publications showing that SRSF6 knockdown alters insulin secretion, direct evidence that these aberrations specifically result from deficient SRSF6-mediated splicing and are not secondary instead to non-splicing activities of this factor (e.g. mRNA stability, export or translation) is missing. The authors argue that this question is beyond the scope of the current manuscript and admit that the functional implications of the observed splicing modifications on beta cell function will require further analyses. Understandably, these would be difficult to perform, especially under the current circumstances. In summary, the manuscript is a useful resource for the definition of the SRSF6 binding site, its position dependent splicing regulation of diabetes susceptibility genes and potential implications on a network of diabetes susceptibility genes, adding an additional layer of regulation to genetic predisposition for diabetes. I reviewed this manuscript with the help of Dr. Jovana Vasiljevic who has expertise in the area of RNA biology and beta cells.

December 15, 2020

RE: Life Science Alliance Manuscript #LSA-2020-00825-TRR

Mrs. Maria Ines Alvelos
ULB Center for Diabetes Research
U.L.B. CP618 Route de Lennik 808
Brussels, Brussels (Anderlecht) 1070
Belgium

Dear Dr. Alvelos,

Thank you for submitting your Research Article entitled "The RNA binding profile of the splicing factor SRSF6 in immortalized human pancreatic β -cells". It is a pleasure to let you know that your manuscript is now accepted for publication in Life Science Alliance. Congratulations on this interesting work.

DISTRIBUTION OF MATERIALS:

Again, congratulations on a very nice paper. I hope you found the review process to be constructive and are pleased with how the manuscript was handled editorially. We look forward to future exciting submissions from your lab.

Sincerely,

Shachi Bhatt, Ph.D.

Executive Editor

Life Science Alliance

<https://www.lsjournal.org/>
